# RAR: Retrieving And Ranking Augmented MLLMs for Visual Recognition

## Abstract

CLIP (Contrastive Language–Image Pre-training) uses contrastive learning from noise image-text pairs to excel at recognizing a wide array of candidates, yet its focus on broad associations hinders the precision in distinguishing subtle differences among fine-grained items. Conversely, Multimodal Large Language Models (MLLMs) excel at classifying fine-grained categories, thanks to their substantial knowledge from pre-training on web-level corpora. However, the performance of MLLMs declines with an increase in category numbers, primarily due to growing complexity and constraints of limited context window size. To synergize the strengths of both approaches and enhance the few-shot/zero-shot recognition abilities for datasets characterized by extensive and fine-grained vocabularies, this paper introduces RAR, a Retrieving And Ranking augmented method for MLLMs. We initially establish a multi-modal retriever based on CLIP to create and store explicit memory for different categories beyond the immediate context window. During inference, RAR retrieves the top-$k$ similar results from the memory and uses MLLMs to rank and make the final predictions. Our proposed approach not only addresses the inherent limitations in fine-grained recognition but also preserves the model's comprehensive knowledge base, significantly boosting accuracy across a range of vision-language recognition tasks. Notably, our approach demonstrates a significant improvement in performance on 5 fine-grained visual recognition benchmarks, 11 few-shot image recognition datasets, and the 2 object detection datasets under the zero-shot recognition setting.

## 1 Introduction

The new material added for the rebuttal discussion is in red.

The CLIP (Contrastive Language–Image Pre-training) (Radford et al., 2021) model and its diverse variants (Sun et al., 2023a; Dong et al., 2023; Li et al., 2023b) provide flexible and robust performance across a wide array of visual-language understanding tasks. Despite its successes, we observe that CLIP's performance begins to wane when faced with datasets characterized by vast vocabularies or fine-grained categories. As shown in the upper left of Fig. 1, the decline is largely attributable to the inherent ambiguity of language descriptions and the challenges posed by synonyms, which can confound the model's ability to distinguish between closely related but distinct classes.

Parallel to these developments, Multi-modal Large Language Models (MLLMs) have emerged as a powerful class of generative models, exemplified by the likes of GPT-4V (OpenAI, 2023) and analogous advancements (Zhu et al., 2023; Liu et al., 2024a; Dai et al., 2023; Peng et al., 2023; Ye et al., 2023; Awadalla et al., 2023; Zhang et al., 2023; Bai et al., 2023; Wang et al., 2023b; Chen et al., 2023). MLLMs, pre-trained on extensive corpora with substantial knowledge, demonstrate remarkable proficiency in identifying fine-grained categories when the total number of candidates remains manageable. Nevertheless, MLLMs' efficacy is similarly compromised in scenarios involving extensive vocabularies and fine-grained categorizations (upper left of Fig. 1).

To address these challenges, we propose augmenting standard MLLMs with our RAR, a retrieving-and-ranking augmented technique. Our RAR enables models to dynamically incorporate external knowledge into the processing and generation workflows. By augmenting MLLMs with external knowledge sources, we address challenges related to language ambiguity, synonym handling, and the limitations imposed by limited context windows when dealing with vast vocabularies. Our method

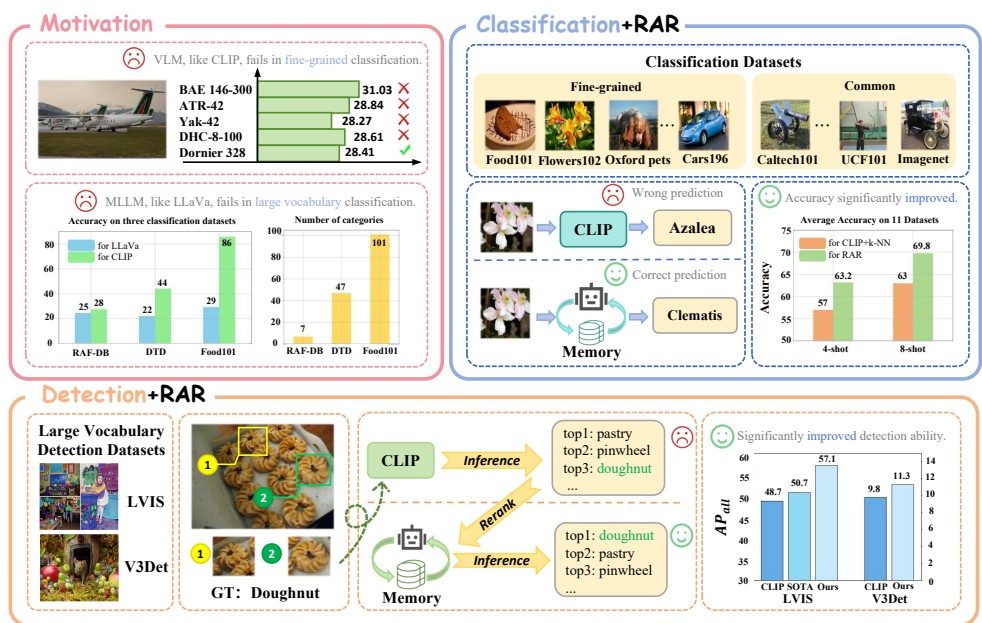

Figure 1: **Upper left**: our motivation about the drawbacks of CLIP and MLLM. Our RAR can seamlessly integrate into MLLMs to improve the few-shot/zero-shot abilities on classification (**upper right**) and detection (**bottom**) datasets.

uses the inherent strength of MLLMs in generalizing from existing knowledge while addressing their limitations in visual recognition. We first construct a multi-modal retriever that creates and stores multimodal embeddings for visual images and text descriptions. As shown in Fig.1, upon receiving an input image at the inference stage, our approach retrieves the top-$k$ class names most similar to the image. Subsequently, the MLLMs rank these retrieved candidate results as the final prediction results. To bolster the MLLMs' ranking performance, we explore fine-tuning with ranking format data or in-context learning examples without training. By integrating our retrieval-augmented design, our approach seeks to bridge the gap between the broad generalization capabilities of MLLMs and the need for precise, fine-grained categorization, offering a path forward that preserves the model's extensive knowledge base while significantly boosting its performance on downstream tasks.

To evaluate our method's efficacy, we conducted benchmarks in three areas: (1) fine-grained visual recognition across 5 benchmarks, (2) few-shot image recognition across 11 datasets, and (3) zero-shot object recognition on 2 object detection datasets with vast vocabularies (*e.g.*, 13204 classes of V3Det (Wang et al., 2023a)). As presented in the right part of Fig. 1, our findings reveal that our approach notably enhances few-shot learning abilities, yielding an average improvement of 6.2% over 11 image classification datasets under the 4-shot setting. Furthermore, our method achieves a 6.4% improvement on the LVIS dataset and a 1.5% gain on the V3Det dataset in zero-shot object recognition performance.

In summary, our key contributions are outlined as follows: (1) We conduct an in-depth analysis of the strengths and weaknesses of VLMs and MLLMs in processing fine-grained datasets. (2) To enhance the fine-grained few-shot and zero-shot perception capabilities of MLLMs, we introduce RAR with a multi-modal retriever and the inference pipeline based on retrieving and ranking. (3) Our RAR can be seamlessly integrated into various MLLMs in a plug-and-play manner. (4) Through rigorous testing across 11 classification datasets and 2 object detection datasets, we demonstrate that our method outperforms baselines on a variety of visual recognition tasks.

## 2 RELATED WORK

**Contrastive Language-Image Pre-training (CLIP)** (Radford et al., 2021) understands images and texts by contrastive learning from a vast amount of visual data paired with natural language descriptions. CLIP has robust capabilities in downstream tasks including image-text retrieval (Yasunaga et al., 2023; Yu et al., 2024; Glass et al., 2022), zero-shot classification (Zhou et al., 2022a; Gao et al.,

2023), and open-vocabulary perception (Gu et al., 2022; Zang et al., 2022; Zhou et al., 2022b). Following CLIP, many subsequent vision-language models (Jia et al., 2021; Li et al., 2022a;b; Zhong et al., 2022; Fang et al., 2023; Dong et al., 2023; Li et al., 2023b; Sun et al., 2023b; Yang et al., 2023b; Lüddecke & Ecker, 2022) are proposed to further improve the vision-language understanding abilities. There are also works done to improve CLIP in zero-shot perception tasks (Subramanian et al., 2022; Shtedritski et al., 2023; Liang et al., 2023; Xu et al., 2023; Yang et al., 2023a). However, simple dot-product between two unimodality features can lead to sub-optimal results for fine-grained classification. In this paper, we demonstrate that CLIP faces challenges in making accurate zero-shot predictions for fine-grained classes, and how our proposed method can effectively re-rank these predictions to improve the accuracy.

**Multimodal Large Language Models** (MLLMs) such as GPT4V (OpenAI, 2023), represent a significant evolution in the landscape of Large Language Models (LLMs) by integrating visual images as input tokens alongside textual information. The integration is facilitated through the use of an additional vision encoder (Radford et al., 2021) and a bridging mechanism (Zhu et al., 2023; Liu et al., 2024a; Dai et al., 2023; Peng et al., 2023; Ye et al., 2023; Awadalla et al., 2023; Zhang et al., 2023; Bai et al., 2023; Wang et al., 2023b; Chen et al., 2023). MLLMs significantly enhance the interaction between humans and AI in more natural and intuitive ways and demonstrate remarkable capabilities in understanding and generating multi-modal content. Despite their prowess, our research uncovers a nuanced limitation: MLLMs tend to underperform in tasks requiring vast vocabularies, where distinguishing subtle differences among different categories is crucial. However, we prove that MLLMs exhibit a strong ability to excel in the re-ranking of top results obtained through vision-language models such as CLIP. Fine-R (Liu et al., 2024b) first delves into leveraging MLLMs for fine-grained perception tasks by prompt design for better descriptions and attributes. We find a new way to prompt it with possible candidates to help screening and achieve better performance.

## 3 METHODOLOGY

We first provide the background information on CLIP, MLLMs, and retrieval-augmentation in LLMs (Sec. 3.1). Then we present the multi-modal retriever (Sec. 3.2) module of RAR and how to apply RAR on downstream tasks via retrieving and ranking (Sec. 3.3).

### 3.1 PRELIMINARIES

**CLIP** is a model combining an image encoder $\Phi_{\text{img}}$ and a text encoder $\Phi_{\text{txt}}$ that uses contrastive learning to understand and align images and text by training on a vast dataset gathered from the web. The core mechanism of CLIP involves mapping an input image $\mathcal{I}$ to its most semantically similar category $c \in \mathcal{C}$:

$$p(y = c|\mathbf{x}) = \arg \max_{c \in \mathcal{C}} \cos(\Phi_{\text{img}}(\mathcal{I}), \Phi_{\text{txt}}(c)),\tag{1}$$

where $y$ represents the predicted category, $\mathcal{C}$ refers to the whole categories list and $\cos(\cdot, \cdot)$ denotes to the cosine similarity.

**Multimodal Large Language Models** such as GPT4V (OpenAI, 2023) learning to generate predictions over sequences of tokens that span both image and text modalities. The MLLM model $f$, parameterized by weights $\theta$, conditioned on the input sequences $\mathbf{x} = (x_1, \ldots, x_{L_{in}})$ of length $L_{in}$, which consist of both text tokens $\mathbf{x}_{\text{txt}}$ and visual tokens $\mathbf{x}_{\text{img}}$. The $\mathbf{x}_{\text{img}}$ are extracted from the input image $\mathcal{I}$ via the image encoder $\Phi_{\text{img}}$. MLLM model forecast a sequence of output tokens $\mathbf{y} = (y_1, \ldots, y_{L_{out}})$ of length $L_{out}$ as follows:

$$p_\theta(\mathbf{y}|\mathbf{x}) = \prod_{l=1}^{L_{out}} p_\theta(y_l|\mathbf{x}, \mathbf{y}_{\leq l-1}) = \prod_{l=1}^{L_{out}} \text{softmax}(f(\mathbf{x}, \mathbf{y}_{\leq l-1}; \theta))_{y_l},\tag{2}$$

where $\mathbf{y}_{\leq l-1} := (y_1, \ldots, y_{l-1})$ refers to the mechanism that predicts the distribution of the next token considering all previously generated tokens.

**Retrieval-Augmentation in Large Language Models** introduces a retrieval module $R$ with the LLM parameterized by $\theta$ for generation. The retrieval module $R$ is designed to process an input sequence $\mathbf{x}$ against an external memory of documents $\mathcal{M}$, efficiently selecting a subset of documents $M \subseteq \mathcal{M}$. The subset $M$ is then fed along with the original input sequence $\mathbf{x}$ into the LLM $\theta$, which uses both the input and the context provided by retrieved results to generate the target output $\mathbf{y}$:

$$p_\theta(\mathbf{y}|\mathbf{x}, M) = \prod_{l=1}^{L_{out}} p_\theta(y_l|\mathbf{x}, M, \mathbf{y}_{\leq l-1}).\tag{3}$$

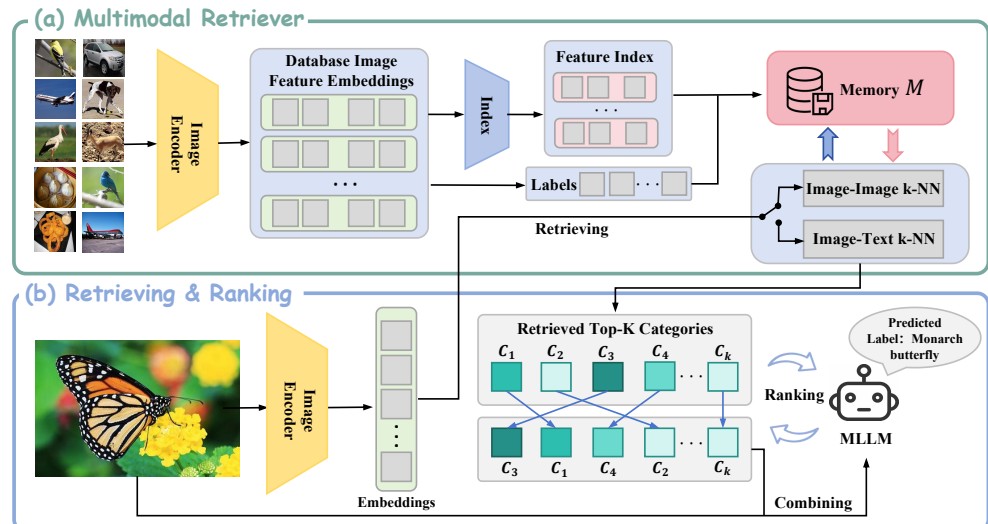

Figure 2: **Pipeline of RAR.** **(a)** We design a **multimodal retriever** that extracts the image or text embeddings and stores embeddings in an external memory $\mathcal{M}$. **(b)** For the inference stage of downstream recognition tasks, we **retrieve** top-$k$ categories from the memory and use MLLMs to refine the retrieved results as the final prediction through **ranking**.

## 3.2 MULTIMODAL RETRIEVER

The multimodal retriever is essentially responsible for querying a large multi-modal external memory or database to find information relevant to the input query or context. In the process of multi-modal retriever, the main challenge lies in efficiently encoding and storing a large volume of images/text embeddings for quick, accurate retrieval. Recognizing the main challenge, as shown in Fig. 2, we have developed a multi-modal retriever that creates and stores multimodal embeddings, with a focus on optimizing retrieval speed through index construction techniques.

**Extracting the Multi-modal Embeddings.** We use the CLIP model discussed in Sec. 3.1 to extract the multi-modal embeddings. Given a data sample $(x_i, c_i)$ from the dataset $\mathcal{D}$ containing the image $x_i$ and class name $c_i$, we use the CLIP image encoder $\Phi_{\text{img}}$ to extract the image embedding $e_{\text{img}} \in \mathbb{R}^d$ and the CLIP text encoder $\Phi_{\text{text}}$ to extract the text embedding $e_{\text{text}} \in \mathbb{R}^d$. The symbol $d$ refers to the feature dimension (*e.g.*, $d = 512$ for CLIP ViT-B/16). The image and text embeddings are stored in the memory $\mathcal{M}$ for retrieval (will discuss in Sec. 3.3). In some zero-shot settings, the image embedding is not available and we merely store the text embedding into the memory.

**Fast Retrieval Optimization.** The brute force search is the common method for designing the retriever, which requires iteration over all vectors in the memory $\mathcal{M}$ to compute similarity scores (*e.g.*, cosine similarity) and subsequently identify the top-$k$ results. Although the brute force method is inherently straightforward, its efficiency markedly diminishes as the dataset escalates to the magnitude of millions of embeddings. To enhance the speed of retrieval, we implement an index system that uses the HNSW(Hierarchical Navigable Small World) algorithm (Malkov & Yashunin, 2018). The adoption of the HNSW methodology facilitates a significant dimensionality reduction, thereby enabling the construction of a more condensed index. Specifically, vectors in a $\mathbb{R}^d$ space of dimension $d$ are transformed into a reduced $\frac{d}{9}$ dimensional space. This reduction in dimensionality plays a pivotal role in enhancing the speed of the retrieval process.

**Pre-processing for Detection Datasets.** In object detection datasets, our methodology for extracting image embeddings $e_{\text{img}}$ is slightly different from the approach discussed previously. As presented in Fig. 3, we apply two additional pre-processing steps: cropping and blurring. Some previous works have proposed similar methods in CLIP like (Yang et al., 2023b; Lüddecke & Ecker, 2022). In the object detection dataset, an image typically contains multiple objects of varying sizes. Some objects may dominate a large portion of the image, whereas others occupy minimal space. Accordingly, our object detection procedure begins with **cropping** the image regions based on proposal bounding box coordinates, subsequently **resizing** the cropped region to a fixed proportion.

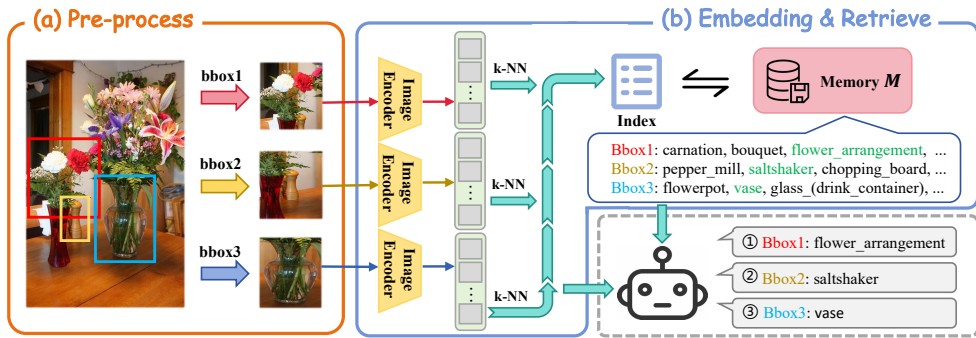

Figure 3: Extending our multimodal retriever to **zero-shot recognition** on object detection datasets such as LVIS (Gupta et al., 2019) and V3Det (Wang et al., 2023a). Compared to the classification datasets, we apply the additional pre-processing techniques such as **cropping** and **resizing** to extract the image embeddings.

Moreover, unlike image classification tasks the objects of interest generally appear large and centrally positioned, the objects within object detection datasets are smaller and their positions more varied. To help the MLLMs understand the objects to be detected, we employ a **blurring** technique on the non-target areas surrounding the objects of interest. The blurring strategy is designed to direct the MLLMs' focus toward the relevant objects, thereby facilitating their identification in object detection tasks.

### 3.3 INFERENCE WITH RETRIEVING AND RANKING

After successfully constructing memory $\mathcal{M}$ by using our multimodal retriever, our next step is to integrate the memory with the retrieval process and use MLLMs to rank the retrieval results and enhance the performance in few-shot/zero-shot perception tasks.

For example, in the inference stage of the few-shot image classification task, we first use the visual encoder $\Phi_{img}$ to process the input image and obtain the corresponding image embedding $\hat{e}$. The visual encoder is identical to the encoder used in our multi-modal retriever. The image embedding $\hat{e}$ is then navigated through the previously constructed memory index and ranked by similarity to identify the top-$k$ related images. Consequently, memory $\mathcal{M}$ yields the names of the retrieved top-$k$ categories, denoted as $\{c_1, c_2, c_3, ..., c_k\}$. The top-k retrieved results serve as a preliminary filter, narrowing down the vast possibilities to those most likely relevant, based on historical data and the semantic closeness of stored labels to the image content.

Since these cropped sub-images are usually small, CLIP's ability to extract features from these low-resolution images is limited. Therefore, in the object detection task, we do not perform image-to-image retrieval but use CLIP's inherent image-text interaction capabilities to conduct image-to-text retrieval. Finally, we also obtain the top-$k$ category information with the highest similarity.

Following the retrieval phase, the retrieved category labels alongside image embedding $\hat{e}$ are integrated and sent to the MLLMs through our ranking prompt. The MLLMs, combining the internal knowledge and the retrieved information, make the final prediction of the image category. Our proposed inference process, using both the retrieval results from our memory bank and subsequent ranking by the MLLM, ensures a more accurate and contextually aware classification prediction. Our design represents a significant advancement in few-shot image classification, enabling our system to handle a wide variety of images and categories with high precision and flexibility.

**Ranking Prompt Format.** Fig. 4 presents our ranking prompt format. The process begins with the prompt `Sort the optional categories: [class a, class b, class c, class d, class e]`, which is dynamically generated to include the top-k class names retrieved from our multimodal retriever. Our method uses the MLLM's ability to rank these retrieved class names. Unlike traditional approaches that might rely solely on the initial retrieval order, our MLLM employs advanced linguistic and semantic analysis to assess the contextual appropriateness of each class name with the input image.

**Fine-tuning for Ranking.** When directly applying MLLMs to ranking the retrieved results, MLLMs may predict some errors such as beyond the given list or occasional misalignment. To fully exploit

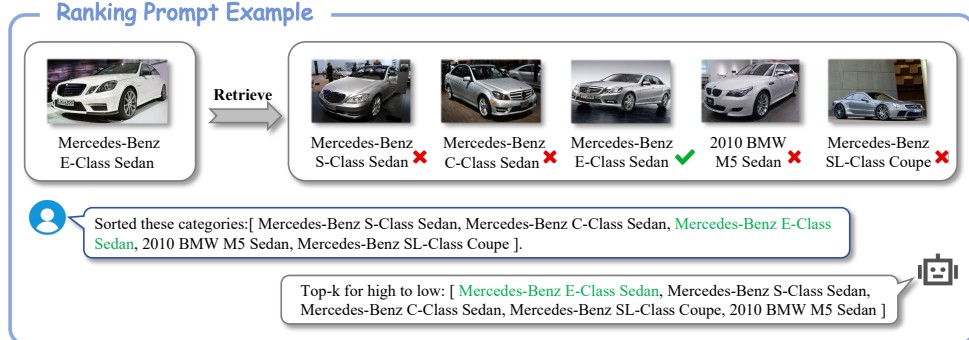

Figure 4: **Ranking Prompt examples** for few-shot image classification. The fine-grained image examples are from Stanford Cars (Krause et al., 2013). We incorporate the initial top-$k$ retrieved results (*e.g.*, $k = 5$) into our ranking prompts and use the MLLMs to rank the retrieved results and make the final prediction.

the ranking potential of MLLMs for downstream tasks, while avoiding the consumption of extensive computational resources for training MLLMs, we selected a small-scale classification dataset to fine-tune the MLLMs. The primary goal of fine-tuning was to enable MLLMs to improve their ranking ability such as following the format of prompts and returning results as required.

To create our fine-tuning data, we use the CLIP image encoder $\Phi_{\text{img}}$ to extract the embeddings of two disjoint subsets of images $\mathcal{D}_a$ and $\mathcal{D}_b$, both drawn from the FGVC-Aircraft dataset. We provide the ablation studies in Sec. 4.5 about using different datasets to construct the fine-tuning data. Our observation reveals that the MLLM demonstrates robustness to the choice of fine-tuning datasets, with only marginal differences in performance outcomes.

For each image in $\mathcal{D}_b$, we apply the $k$-NN clustering algorithm to find the top 20 most similar images in $\mathcal{D}_a$ including their categories. Afterward, we select 16 sets from these 20 images, each set comprising $k$ images, and retain those groups that contain images of the same category as $\mathcal{D}_b$. We then shuffled the category labels for these sets. Using the prompts shown in Fig. 4, we create a dataset comprising roughly 30,000 entries, with the original sequence of categories serving as the ground-truth label. In summary, we build the fine-tuning data aiming to bolster the MLLM's ranking performance.

**In-Context Learning for Ranking.** In-context learning presents a valuable alternative to fine-tuning with ranking examples, particularly due to its flexibility and lower requirement for specialized data preparation. While fine-tuning with ranking examples has proven to be highly effective, it necessitates a substantial amount of curated data and computational resources for training. In contrast, in-context learning uses the model's existing knowledge by providing it with specific examples directly within the input prompt, guiding the model to understand and execute the task of ranking without the need for explicit re-training. Here we elaborate on the application of in-context learning with MLLMs to rank the retrieved results. To effectively guide the MLLMs in comprehending the ranking task, we use the prompt format similar to Fig. 4 and integrate a specific ranking example into the prompts. Please refer to the Sec. B for our structured in-context learning prompt. Please refer to Sec. 4.5 for the ablation studies of discussing the difference between using fine-tuning or in-context learning for ranking.

## 4 EXPERIMENTS

In this section, we present our experiment step (Sec. 4.1) and conduct experiments on different tasks such as fine-grained visual recognition (Sec. 4.2), few-shot image recognition (Sec. 4.3) and zero-shot object recognition (Sec. 4.4). We also provide the ablation studies about our design choices (Sec. 4.5).

### 4.1 EXPERIMENTAL SETUP

**Datasets and Evaluation Metrics.** We follow previous work (Liu et al., 2024b) to choose 5 datasets for **fine-grained visual recognition** (Bird-200 (Wah et al., 2011), Cars-196 (Krause et al., 2013), Dog-120 (Khosla et al., 2011), Flower-102 (Nilsback & Zisserman, 2008), and Pet-37 (Parkhi et al.,

Table 1: Fine-grained visual recognition across 5 datasets. We follow (Liu et al., 2024b) to report the averaged clustering accuracy (cACC, %) and semantic similarity accuracy (sACC, %) results over 10 runs. The best and second-best results are colored Green and Red , respectively.

| | Bird-200 | | Car-196 | | Dog-120 | | Flower-102 | | Pet-37 | | Average | |
| | cACC | sACC | cACC | sACC | cACC | sACC | cACC | sACC | cACC | sACC | cACC | sACC |
|---|---|---|---|---|---|---|---|---|---|---|---|---|
| WordNet+CLIP | 39.3 | 57.7 | 18.3 | 33.3 | **53.9** | **70.6** | 42.1 | 49.8 | 55.4 | 61.9 | 41.8 | 54.7 |
| BLIP-2 | 30.9 | 56.8 | 43.1 | 57.9 | 39.0 | 58.6 | 61.9 | **59.1** | 61.3 | 60.5 | 47.2 | 58.6 |
| CaSED | 25.6 | 50.1 | 26.9 | 41.4 | 38.0 | 55.9 | **67.2** | 52.3 | 60.9 | 63.6 | 43.7 | 52.6 |
| FineR | 51.1 | **69.5** | 49.2 | 63.5 | 48.1 | 64.9 | 63.8 | 51.3 | 72.9 | 72.4 | 57.0 | 64.3 |
| RAR (Ours) | **51.6** | **69.5** | **53.2** | **63.6** | 50.0 | 65.2 | 63.7 | 53.2 | **74.1** | **74.8** | **58.5** | **65.3** |

2012)) and report the clustering accuracy (cACC) and semantic similarity accuracy (sACC) as evaluation metrics.

For **few-shot image recognition**, we select 11 datasets including general objects (ImageNet (Deng et al., 2009), Caltech101 (Fei-Fei et al., 2004)), textual (DTD (Cimpoi et al., 2014)), scene objects (SUN397 (Xiao et al., 2010)), satellite images (EuroSAT (Helber et al., 2019)), facial expressions (RAF-DB (Li et al., 2017)), car types (Stanford Cars (Krause et al., 2013)) and fine-grained datasets (FGVC-Aircraft (Maji et al., 2013), Oxford Flowers (Nilsback & Zisserman, 2008), Food101 (Nilsback & Zisserman, 2008) and Oxford Pets (Parkhi et al., 2012)). We report the top-1 accuracy (%) for all these classification datasets.

Additionally, we also select two benchmarks for our **zero-shot object recognition** setting: (1) The LVIS(Gupta et al., 2019) dataset that encompasses over 164,000 images and 1,203 categories. We report the $AP_r$, $AP_c$, $AP_f$, and $AP_{all}$ metrics for rare, common, frequent, and all categories. (2) V3Det (Wang et al., 2023a) dataset encompasses an immense number of 13204 categories of real-world images. For V3Det, we report the standard mAP metric of the object detection task.

**Implementation Details.** We employ a frozen CLIP ViT B/16 model as the visual encoder $\Phi_{img}$ to encode the input images and extract the corresponding image embeddings. For the retrieval process, we search the stored embeddings in memory $\mathcal{M}$ using the HNSW algorithm (Malkov & Yashunin, 2018). We use $k = 5$ for the top-$k$ results, with a solo exception $k = 4$ in the 4-shot few-shot setting. To improve the ranking ability of MLLMs, we prepare 30k fine-tuning data from the FGVC-Aircraft dataset. In the fine-tuning process, we train the model with one epoch with a learning rate of $1e^{-5}$ on our fine-tuning data and subsequently evaluate the performance across additional datasets. We present the ablation studies about the hyper-parameters such as the value of $k$ and the fine-tuning data source in the Sec. 4.5.

### 4.2 FINE-GRAINED VISUAL RECOGNITION

We first evaluate our RAR on the *fine-grained visual recognition* setting defined in previous work (Liu et al., 2024b). We use only 3 unlabelled images per category to build our memory $\mathcal{M}$ for retrieving. Please refer to Sec. C for more implementation details.

**Baselines.** We follow (Liu et al., 2024b) to select four representative methods as our baselines to compare with: WordNet (Miller, 1995)+CLIP, BLIP-2 (Li et al., 2023a), CaSED (Conti et al., 2024), and FineR (Liu et al., 2024b).

**Averaged Results over 5 Datasets.** Tab. 1 summarizes the results and our RAR achieves the top performance on both the cACC (58.5%) and sACC (65.3%) metrics. The WordNet+CLIP and CaSED baselines rely solely on CLIP for class name retrieval, yet often yield inaccurate predictions. In contrast, our method adds the additional ranking process with MLLMs, which increases the likelihood of correctly predicting those accurate yet initially lower-ranked candidates and thereby boosts performance. Besides, FineR uses MLLM (*e.g.*, BLIP-2) for fine-grained recognition via multi-round questioning-answering processes, which may demand more computational resources and struggle to scale efficiently with large vocabulary datasets. Conversely, our approach first retrieves candidates and then lets MLLMs make predictions on the candidates, optimizing both accuracy and efficiency.

We can observe that RAR did not achieve SOTA results on Dog-120 and Flower-102. This is because some baselines use exhaustive knowledge bases on specialized datasets: WordNet covers all ground-truth Dog-120 categories, and CaSED includes 101 of 102 ground-truth Flower-102 categories. As

Table 2: Few-shot image classification across 11 datasets. We report the top-1 accuracy (%) under the 4-shot and 8-shot settings. Here our RAR uses the LLaVA1.5 (Liu et al., 2023) as the MLLM to rank the retrieved results. The symbol '-' denotes to the LLaVA model fails to make the predictions due to the limited window size.

| Method | Common | | | | | | | Fine-Grained | | | | |
|---|---|---|---|---|---|---|---|---|---|---|---|---|
| | ImageNet | Caltech101 | RAF-DB | SUN397 | EuroSAT | DTD | UCF-101 | Flower102 | StanfordCars | Food101 | OxfordPets | Average |
| **4-shot** | | | | | | | | | | | | |
| CLIP+KNN | 42.1 | 87.9 | 14.2 | 51.4 | 67.6 | 47.5 | 64.6 | **84.5** | 49.2 | 62.6 | 55.6 | 57.0 |
| LLaVA1.5 Finetuning | - | 88.4 | 24.9 | - | 48.2 | 46.6 | 58.9 | 13.2 | - | 66.4 | 28.9 | - |
| RAR (LLaVA1.5) | **51.0** | **92.1** | **27.7** | **58.8** | **74.8** | **53.9** | **69.6** | 80.4 | **54.4** | **71.4** | **60.9** | **63.2** |
| Δ | +9.9 | +4.2 | +13.5 | +7.4 | +7.2 | +6.4 | +5.0 | -4.1 | +5.2 | +8.8 | +5.3 | +6.2 |
| **8-shot** | | | | | | | | | | | | |
| CLIP+KNN | 47.6 | 90.6 | 28.2 | 56.8 | 72.8 | 53.2 | 68.3 | **89.5** | 56.1 | 68.3 | 61.8 | 63.0 |
| LLaVA1.5 Finetuning | - | 92.1 | 24.9 | - | 48.2 | 54.7 | 66.5 | 30.1 | - | 72.5 | 46.1 | - |
| RAR (LLaVA1.5) | **56.5** | **93.5** | **46.9** | **63.4** | **81.5** | **59.3** | **74.3** | 87.3 | **61.2** | **76.6** | **67.7** | **69.8** |
| Δ | +8.9 | +2.9 | +18.7 | +6.6 | +8.7 | +6.1 | +6.0 | -2.2 | +5.1 | +8.3 | +5.9 | +6.8 |

discussed in FineR (Liu et al., 2024b), this leads to biased high performance. Moreover, BLIP-2 uses a more powerful 11B Flan-T5xxl encoder. RAR does not use these exhaustive knowledge bases but still achieves the best results on majority fine-grained datasets (average +16.7%/+11.3%/+14.8% gains over WordNet/BLIP-2/CaSED), which demonstrate RAR is effective and general.

## 4.3 FEW-SHOT IMAGE RECOGNITION

The few-shot setting aims to enable a model to recognize new objects with only a few examples for each new category. Few-shot learning faces substantial challenges when applied to fine-grained datasets, which consist of numerous highly similar classes yet are accompanied by only a minimal amount of training data.

**Baselines.** For *few-shot image recognition*, we introduce two baselines including CLIP and MLLMs. The first is the CLIP (Radford et al., 2021) model combined with $k$-NN to retrieve predictions based on few-shot examples. The second is the LLaVA model directly fine-tuning with LoRA (Hu et al., 2021) on few-shot examples.

**Averaged Results on 11 Datasets.** Tab. 2 summarizes the few-shot results on 11 datasets, including 4 fine-grained datasets. Compared to the CLIP initial retrieval results (top row), our RAR (third row) with ranking facilitates a notable increase in classification accuracy. On average, our approach boosts the top-1 accuracy from 57.0 to 63.2 (%) on the 4-shot setting, and from 63.0 to 69.8 (%) on the 8-shot setting. Such improvements illustrate the ranking process of MLLMs effectively uses a nuanced understanding of context and detail to better align predictions with ground truth. Additionally, we observe that LLaVA1.5 + fine-tuning (second row) baseline underperforms in datasets with large vocabularies such as ImageNet due to the constraint of LLMs' context window. Thanks to the retrieved candidates, our RAR works for datasets with a vast of categories and is a potent tool in refining classification decisions, proving particularly useful in handling the diverse and challenging landscape of image classification tasks.

## 4.4 ZERO-SHOT OBJECT RECOGNITION

Given the pre-existing object proposals such as ground-truth box annotations, the zero-shot object recognition task measures the model's capability of aligning regions with textual class descriptions.

**Baselines.** We select two representative papers CLIP (Radford et al., 2021) and RegionCLIP (Zhong et al., 2022) and report their performances as the baseline results. Besides, we apply our method on a range of cutting-edge open-source MLLMs, including LLaVA1.5 (Liu et al., 2023), QWen-VL (Bai et al., 2023) and InternLM-XC2 (Dong et al., 2024).

**Main Results on LVIS.** Tab. 3 presents the results that reveal notable metrics improvements when applying our RAR. Specifically, when combing with the recent InternLM-XC2 (Dong et al., 2024) model, our approach yielded an 8.4 (%) point increase over the CLIP baseline and a 6.4 (%) enhance-

Table 3: Zero-shot object recognition on LVIS (Gupta et al., 2019) v1.0 *validation* set.

| | $AP_r$ | $AP_c$ | $AP_f$ | $AP_{all}$ |
|---|---|---|---|---|
| CLIP w/ box | 40.6 | 53.1 | 59.2 | 48.7 |
| CLIP w/ mask | 40.8 | 53.5 | **59.6** | 49.2 |
| RegionCLIP | 50.1 | 50.1 | 51.7 | 50.7 |
| RAR (LLaVA1.5) | 58.7 | 57.9 | 54.4 | 56.2 |
| $\Delta$ | +8.6 | +7.8 | +2.7 | +5.5 |
| RAR (Qwen-VL) | 59.6 | 57.5 | 53.7 | 56.4 |
| $\Delta$ | +9.5 | +7.4 | +2.0 | +5.7 |
| RAR (InternLM-XC2) | **60.2** | **58.0** | 54.3 | **57.1** |
| $\Delta$ | +10.1 | +7.9 | +2.6 | +6.4 |

Table 4: Zero-shot object recognition on V3Det (Wang et al., 2023a) *validation* set with 13,204 categories.

| | $AP_s$ | $AP_m$ | $AP_l$ | $AP_{all}$ |
|---|---|---|---|---|
| CLIP w/ box | 7.2 | 12.9 | 12.8 | 9.8 |
| RAR (LLaVA1.5) | 9.9 | **13.2** | 13.9 | 11.1 |
| $\Delta$ | +2.7 | +0.3 | +1.1 | +1.3 |
| RAR (Qwen-VL) | 9.6 | 12.7 | 13.7 | 10.8 |
| $\Delta$ | +2.4 | -0.2 | +0.9 | +1.0 |
| RAR (InternLM-XC2) | **10.1** | 13.1 | **14.5** | **11.3** |
| $\Delta$ | +2.9 | +0.2 | +1.7 | +1.5 |

Figure 5: **Visualization of the ranking examples** for zero-shot object recognition on LVIS (Gupta et al., 2019) *validation* set. Given the top retrieved predictions, our RAR uses MLLMs to select the correct class names accurately.

ment relative to RegionCLIP (Zhong et al., 2022). These advancements underscore the efficacy of using an external memory for retrieval assistance coupled with the ranking prowess of MLLMs.

**Comparison with Rare Classes Results ($AP_r$).** We find an interesting observation from the experimental results presented in Tab. 3. For the CLIP model, we observe a progressive increase in performance from $AP_r$ through $AP_c$ to $AP_f$, which indicates a gradation in precision across varying class frequencies. However, employing our method yields a different trend, where *the peak performance is achieved on $AP_r$*, surpassing the CLIP model by as much as 19.6 percentage points. This significant leap in performance suggests a substantial advantage of our method when it comes to rare categories. The integration of our RAR to MLLMs plays a pivotal role here, as it demonstrates a heightened ability to discriminate among the rare classes. Our observation could be attributed to the fact that our retrieving and reranking mechanism effectively pools relevant information from the external memory, providing the MLLMs with a richer context for rare class identification. Moreover, the ranking capability of MLLMs ensures that even the lesser-represented classes receive adequate attention during the classification process. Our RAR achieves a robust enhancement in the model's ability to discern and accurately classify objects that are infrequently encountered, addressing one of the significant challenges in long-tailed distribution datasets.

**Main Results on V3Det.** To further test the effectiveness of using MLLMs for ranking in scenarios with an extremely large number of fine-grained categories, we conducted additional experiments on V3Det (Wang et al., 2023b). The experimental results in Tab. 4 reveal that our RAR has achieved a commendable improvement in performance, surpassing the CLIP baseline by 1.5 percentage points in overall average precision ($AP_{all}$) with InternLM-XC2. Such an improvement is particularly significant given the complexity of the V3Det dataset, which presents challenging 13,204 distinct classes. The MLLMs, with the aid of our retrieving and ranking mechanisms, have once again demonstrated their robust performance in the domain of object detection datasets. Using our retrieval-augmented approach allows MLLMs to navigate the extensive and fine-grained category landscape of V3Det effectively.

**Qualitative Results.** Fig. 5 presents the visualization results about ranking examples of our approach on LVIS *validation* set. The CLIP&$K$-NN approach provides an extensive list of object predictions, albeit with the caveat that the most accurate label might not always emerge as the top-1 choice. The incorporation of MLLMs in our RAR significantly streamlines the prediction process,

Table 5: **Ablation studies** about (1) using different datasets for fine-tuning and (2) fine-tuning *vs* in-context learning. The symbols 'F' and 'S' stand for fine-tuning on the FGVC-Aircraft or Stanford-Cars datasets.

| Method | Strategy | | Common | | | | | | | Fine-Grained | | | |
|---|---|---|---|---|---|---|---|---|---|---|---|---|---|
| | Fine-tune | In-Context | ImageNet | Caltech101 | RAF-DB | SUN397 | EuroSAT | DTD | UCF101 | Flower102 | Food101 | OxfordPets | Average |
| RAR | F | ✗ | **75.8** | **95.5** | **66.0** | 72.7 | **90.7** | **72.5** | **81.4** | **97.5** | 88.1 | **87.2** | **82.7** |
| | S | ✗ | 75.3 | 94.9 | 65.1 | **73.1** | 88.1 | 71.0 | 81.1 | 95.8 | **88.3** | 87.0 | 82.0 |
| (QWen-VL) | ✗ | ✓ | 72.0 | 93.4 | 63.6 | 65.6 | 86.2 | 66.8 | 76.5 | 95.6 | 84.7 | 82.3 | 78.7 |
| RAR | F | ✗ | **71.5** | 94.4 | **72.7** | **69.7** | 91.7 | **69.9** | **77.6** | 93.2 | 83.9 | 79.3 | **80.4** |
| | S | ✗ | **71.5** | **94.7** | 71.2 | **69.7** | 90.3 | **69.9** | 77.5 | 92.0 | 83.6 | **79.7** | 80.0 |
| (InternLM-XC2) | ✗ | ✓ | 69.2 | 94.1 | 66.0 | **69.7** | **91.8** | 68.9 | 66.1 | **95.7** | **85.7** | 79.2 | 78.6 |

yielding more precise and relevant object labels. The visualization results demonstrate that our RAR meets the need for fine-grained and large vocabulary recognition.

## 4.5 ABLATION EXPERIMENTS

**Different Fine-tuning data.** We study the importance of using different fine-tuning datasets for ranking. We select two representative datasets: FGVC-Aircraft and Stanford-Cars as the data sources for constructing the fine-tuning data. Our selection is motivated by their diverse characteristics and relevance in visual recognition tasks, providing a comprehensive basis for fine-tuning. Subsequently, we fine-tune the RAR with different MLLMs (QWen-VL and InternLM-XC2) on these two datasets, aiming to investigate how different data sources influence performance. To thoroughly assess the impact of using different fine-tuning datasets, we evaluate the fine-tuned RAR across a diverse set of 10 additional datasets.

Tab. 5 presents the results. We observe that RAR is not sensitive to changes in the fine-tuning dataset for ranking, thereby confirming its viability as a generalizable and reliable method for enhancing the performance of MLLMs. The consistency in results, irrespective of the fine-tuning data source, underlines the robustness of our fine-tuning strategy. Despite these minor variations, the overall performance of using FGVC-Aircrafts (82.7%, top row) is higher than using StanfordCars (82.0%, second row) for QWen-VL, and we observe the same trend for InternLM-XC2. Based on our findings, we adopt the FGVC-Aircraft dataset as our preferred choice for fine-tuning.

**Fine-tuning *vs* In-Context Learning.** We validate the effectiveness of fine-tuning the MLLM or just in-context learning (training-free) for ranking. The results are illustrated in Tab. 5. We select two distinct groups for comparison. The first group (top and fourth rows) involves models that are fine-tuned using the FGVC-Aircraft dataset, while the second group (third and bottom rows) consists of models with in-context learning prompts for ranking. The results show a consistent improvement in accuracy for the fine-tuned model across almost all datasets for both QWen-VL and InternLM-XC2. The notable enhancement in performance across a diverse range of datasets highlights the efficacy of our fine-tuning strategy. The results substantiate that fine-tuning the MLLM with target datasets like FGVC-Aircraft significantly bolsters the model's ranking capabilities.

## 5 CONCLUSION

In this paper, we highlight the potential of combining retrieving and ranking with multi-modal large language models to revolutionize perception tasks such as fine-grained recognition, zero-shot image recognition, and few-shot object recognition. Motivated by the limited zero-shot/few-shot of CLIP and MLLMs on fine-grained datasets, our RAR designs the pipeline that uses MLLM to rank the retrieved results. Our proposed approach can be seamlessly integrated into various MLLMs for real-world applications where the variety and volume of categories continuously expand. Our method opens up new avenues for research in augmenting the MLLM's abilities with the retrieving-augmented solution and could be beneficial for other tasks such as reasoning and generation in future works.

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

# Appendix

In this appendix, we provide a series of detailed supporting materials to aid in a deeper understanding of our work. Firstly, in Sec. A, we introduce the fourteen image classification datasets involved in our experiments, including seven common datasets and seven fine-grained datasets, as well as two large-scale vocabulary detection datasets. Following that, in Sec. B, we provide detailed information on the prompts used in our RAR, as well as the prompts used in corresponding ablation studies. In Sec. C, we supplement details on the structure and experimental aspects of RAR, dividing the content into three sections: Fine-Grained Visual Recognition, Few-Shot Image Classification, and Zero-Shot Region Recognition.

## A  DATASET STATISTICS

In this section, we delve deeper into the specifics of the fourteen classification and two detection datasets employed in our research. The classification datasets encompass a wide range, from general categories that cover a broad spectrum of common objects to fine-grained types that focus on more specific, detailed distinctions within a particular category. The detection datasets, on the other hand, are extensive, encompassing tens of thousands of object categories. These datasets are designed to challenge the model's ability to identify and categorize objects from a vast array of possible classes. The long-tail nature of these datasets poses a significant challenge for our RAR model.

### A.1  CLASSIFICATION DATASETS

In the experimental part, we use a total of fourteen image classification datasets, including seven fine-grained classification datasets and seven common classification datasets. Fine-grained image classification datasets include: Bird-200 (Wah et al., 2011), Stanford Cars (Krause et al., 2013), Dog-120 (Khosla et al., 2011), Oxford Flowers (Nilsback & Zisserman, 2008), Oxford Pets (Parkhi et al., 2012), FGVC-Aircraft (Maji et al., 2013), and Food101 (Nilsback & Zisserman, 2008). Common image classification datasets include: ImageNet (Deng et al., 2009), Caltech101 (Fei-Fei et al., 2004), RAF-DB (Li et al., 2017), Sun397 (Xiao et al., 2010), Eurosat (Helber et al., 2019), DTD (Cimpoi et al., 2014), and UCF-101 (Soomro et al., 2012). We present all the utilized datasets in Fig. 6. And in Tab. 6, we list the statistics and sources of these datasets in detail.

In our fine-grained visual recognition experiments, we employed the following datasets: Bird-200, Stanford Cars, Dog-120, Flowers-102, and Oxford pets. In each dataset, we selected 3 images from the training set to construct our memory and conducted tests on the corresponding validation sets. In our few-shot image classification experiments, we used the FGVC-Aircraft dataset to build fine-tune data and tested our RAR model across eleven classification datasets: Stanford Cars, Flower-102, Oxford Pets, Food101, ImageNet, Caltech101, RAF-DB, Sun397, Eurosat, DTD, and UCF-101. We selected either 4 or 8 images from the training set of each dataset to place into memory, corresponding to 4-shot and 8-shot settings, respectively, and conducted tests across all validation sets.

### A.2  DETECTION DATASETS

In our Zero-Shot Region Recognition experiments, we utilized two large-scale vocabulary detection datasets, namely LVIS and V3Det. The LVIS dataset, developed by Facebook AI researchers, stands out with its extensive coverage, including 164,000 images and about 2,000,000 high-quality instance segmentation annotations that span over 1,000 object classes. This dataset is particularly notable for its long-tail distribution, which means it includes a large number of infrequent or rare object classes in addition to the common ones. This diversity challenges our model to recognize and differentiate between a wide array of objects, including those that are less common and hence more challenging to identify accurately.

The V3Det dataset complements LVIS by offering an even broader scope. With its 245,000 images distributed across an impressive 13,204 categories, V3Det brings an unprecedented level of diversity to the table. The dataset includes 1,753,000 meticulously annotated bounding boxes, making it an invaluable resource for developing and testing detection algorithms capable of handling a wide

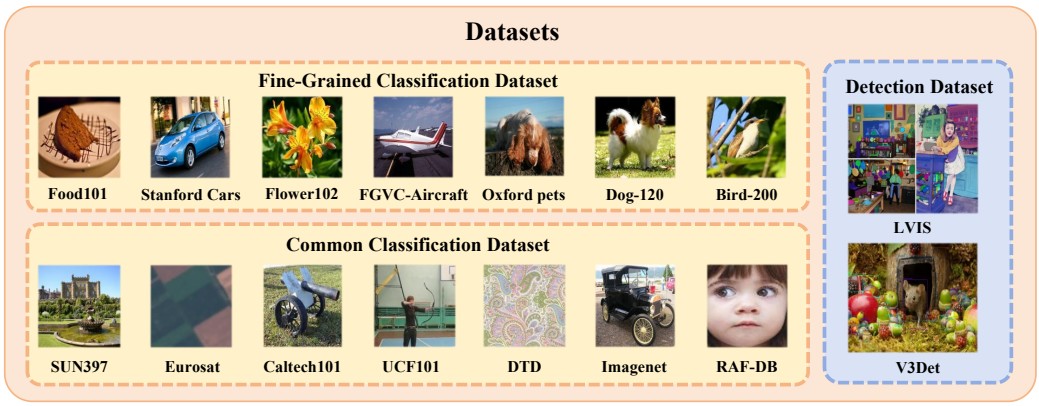

Figure 6: **Datsets** used in our experiments. We select 14 classification datasets (7 fine-grained and 7 common) and 2 object detection datasets as our benchmarks.

Table 6: Statistics for the classification and detection datasets used in our three settings: fine-grained visual recognition, few-shot image recognition, and zero-shot region recognition.

| Settings | Dataset | Categories | Evaluation Metrics | Source link |
|---|---|---|---|---|
| Fine-Grained Visual Recog. | Bird-200 | 200 | cACC, sACC | Bird website |
| | Car-196 | 196 | cACC, sACC | Kaggle |
| | Dog-120 | 120 | cACC, sACC | Tensorflow |
| | Flower-102 | 102 | cACC, sACC | Tensorflow |
| | Pet-37 | 37 | cACC, sACC | Tensorflow |
| Few-Shot Image Recog. | RAF-DB | 7 | Accuracy | RAF-DB website |
| | Eurosat | 10 | Accuracy | Tensorflow |
| | DTD | 47 | Accuracy | Tensorflow |
| | FGVC Aircraft | 100 | Accuracy | FGVC website |
| | Caltech101 | 101 | Accuracy | Tensorflow |
| | Food101 | 101 | Accuracy | Tensorflow |
| | UCF-101 | 101 | Accuracy | Tensorflow |
| | SUN397 | 397 | Accuracy | Tensorflow |
| | ImageNet | 1000 | Accuracy | Tensorflow |
| Zero-Shot Region Recog. | LVIS | 1203 | mAP | LVIS website |
| | V3Det | 13204 | mAP | Github |

variety of object types. Its large number of categories ensures that the dataset has a comprehensive representation of the visual world, making it an ideal testing ground for our Zero-Shot Region Recognition experiments.

## B PROMPT FORMATS

In this section, we delve into the detailed design of our prompts. We have crafted distinct prompts for various tasks to test the capabilities of the baseline model and our RAR model in visual recognition.

In our RAR pipeline, the prompt primarily serves to merge the input image with the category information retrieved from memory. It guides MLLMs to rank the retrieved candidate object categories based on similarity. Our prompt format is as follows:

```
Please play the role of a classification expert, and sort the provided
categories from high to low according to the {top-k} similarity with the
input image. Here are the optional categories:{categories}.
```

Here, '{top-k}' is replaced with the number of categories input. And '{categories}' is replaced with the top-k categories retrieved from memory.

Additionally, to assess the visual recognition and ranking capabilities of MLLMs themselves, we have prepared a prompt with examples to serve as input for the model. Our structured in-context learning prompt is as follows:

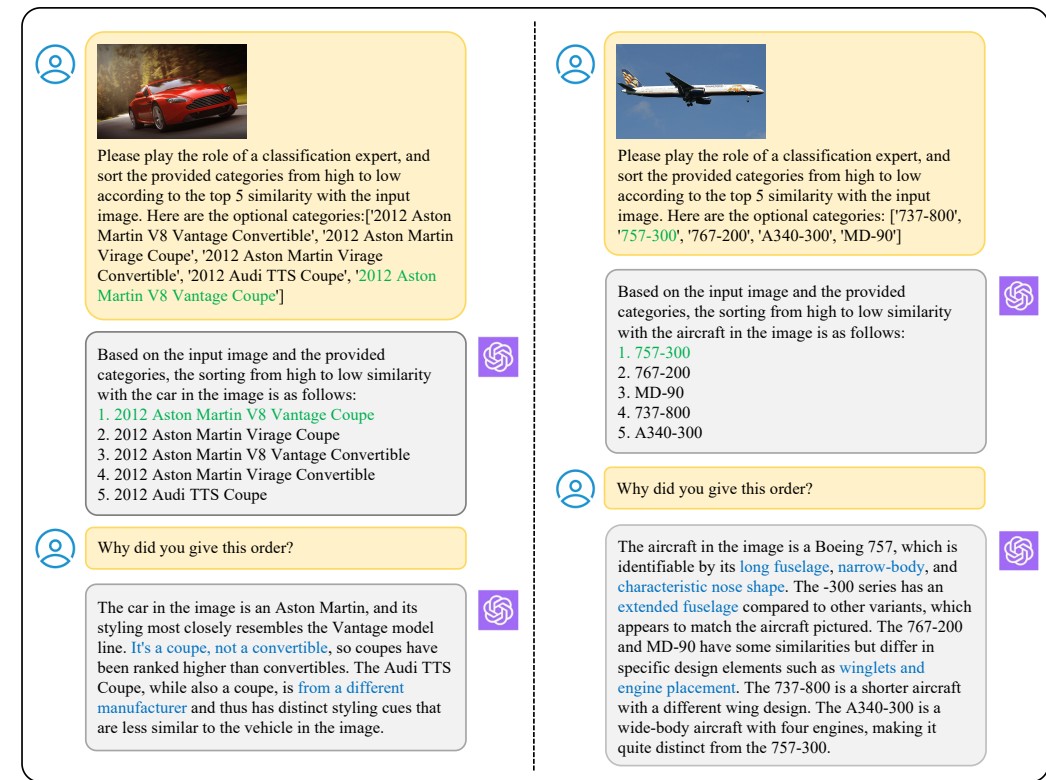

Figure 7: **GPT4V Example** for Stanford Cars and FGVC Aircraft. **Green** for ground truth, **blue** for characteristics analyzed by GPT-4V.

```
Please play the role of a classification expert, and sort the provided
categories from high to low according to the top 5 similarity with the
input image. Here are the optional categories:{categories}.Your answer
should follow the following format, like:['category A', 'category B',
'category C', 'category D', 'category E']. Only choose five categories,
and no further information.
```

When testing the RAR pipeline with MLLMs, '{categories}' is replaced with all the category names of each dataset.

## C  MORE IMPLEMENTED DETAILS AND EXPERIMENTS

### C.1  FINE-GRAINED VISUAL RECOGNITION

In the fine-grained visual recognition section, we first evaluate our RAR on the setting defined in previous work (Liu et al., 2024b). For each category in the five datasets, we select three unlabeled images to form a 3-shot setting. Then, we extract embeddings using the CLIP B/16 model and store them in memory. The labels for each image correspond to the predictions in [31]. We then test the validation set using the RAR pipeline and measure the results with Clustering Accuracy (cACC) and Semantic Similarity (sACC).

**Evaluation Metrics.** In the fine-grained visual recognition section, we use two synergistic metrics: Clustering Accuracy (cACC) and Semantic Similarity (sACC) to evaluate our method, following (Liu et al., 2024b). Clustering Accuracy (cACC) mainly assesses the accuracy of clustering images within the same category, without considering the semantic relatedness of category labels. Complementing this, Semantic Similarity (sACC) measures the similarity between the names of categories in the clusters and the ground truth.

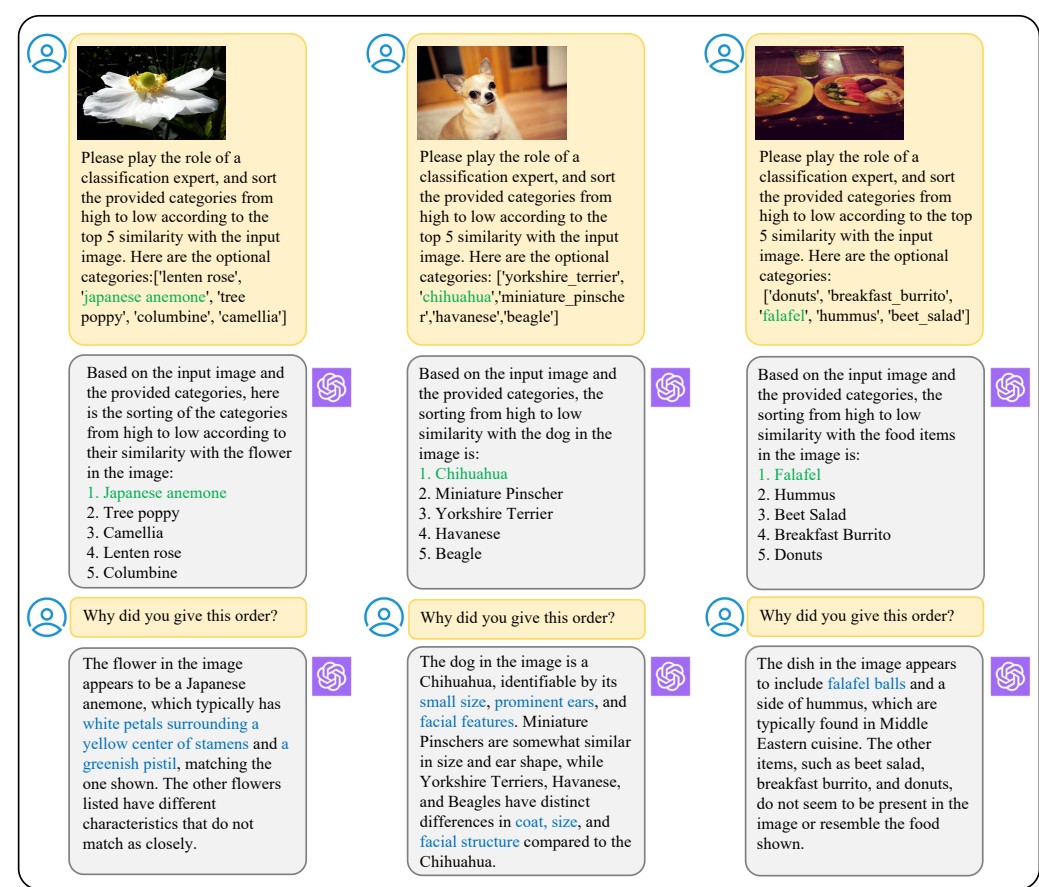

Figure 8: **GPT4V Example** for Flowers102, Pets37 and Food101. **Green** for ground truth, **blue** for characteristics analyzed by GPT-4V.

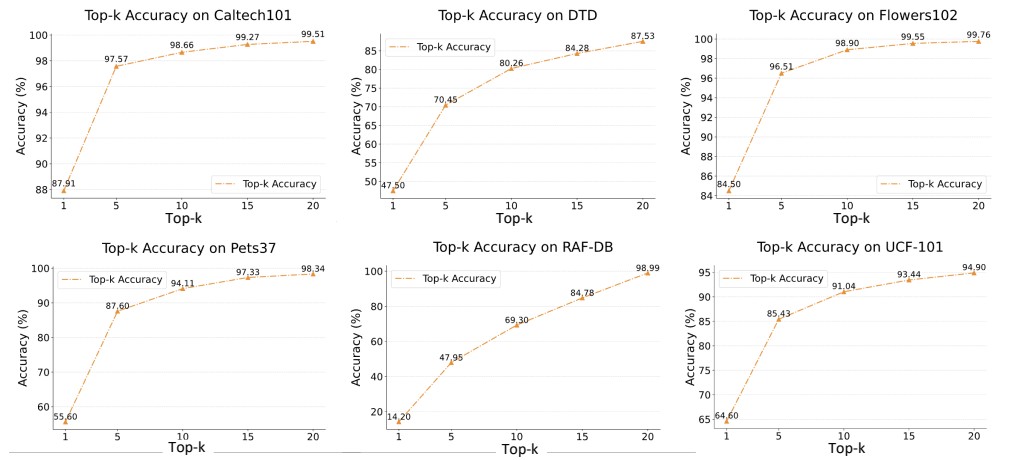

Figure 9: **Evaluation on CLIP+KNN** for Caltech101, Flowers102, RAF-DB, Pets37, DTD and UCF101. We report the top-1, 5, 10, 15, 20 accuracy (%) under the 4-shot settings.

## C.2 FEW-SHOT IMAGE CLASSIFICATION

In this section, we delve deeper into some intriguing observations and motivations behind our study. Additionally, we have included an array of expanded test results in this part, encompassing clas-

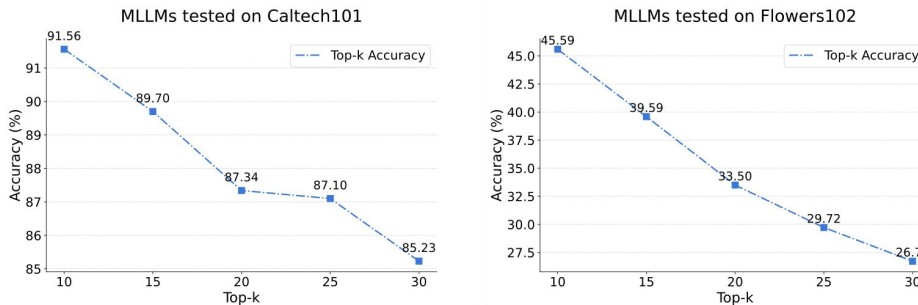

Figure 10: **Evaluation on MLLMs** for Caltech101, Flowers102. We report the test results using 10, 15, 20, 25, and 30 category names as inputs.

Table 7: Few-shot image classification across 11 datasets. We report the top-1 accuracy (%) under the 1-shot, 2-shot, 4-shot, 8-shot and 16-shot settings. The CLIP+KNN method does not utilize the text encoder of CLIP. Instead, we employ the visual encoder to extract image features, and then apply the KNN algorithm to these features. Here our RAR uses the LLaVA1.5 (Liu et al., 2023) as the MLLM to rank the retrieved results. The symbol '-' denotes to the LLaVA model fails to make the predictions due to the limited window size.

| Method | Common | | | | | | | Fine-Grained | | | | |
|---|---|---|---|---|---|---|---|---|---|---|---|---|
| | ImageNet | Caltech101 | RAF-DB | SUN397 | EuroSAT | DTD | UCF-101 | Flower102 | StanfordCars | Food101 | OxfordPets | Average |
| **1-shot** | | | | | | | | | | | | |
| CLIP+KNN | 29.2 | 75.9 | 11.3 | 37.7 | 53.9 | 35.1 | 47.8 | **66.7** | 32.6 | 45.3 | 41.3 | 43.3 |
| LLaVA1.5 Finetuning | - | 84.1 | 24.9 | - | 48.2 | 22.3 | 35.4 | 4.59 | - | 39.2 | 16.3 | - |
| RAR (LLaVA1.5) | **40.3** | **85.2** | **34.8** | **46.5** | **62.4** | **38.1** | **57.4** | 50.4 | **38.3** | **57.6** | **47.0** | **50.7** |
| Δ | +10.5 | +9.3 | +23.5 | +8.8 | +8.5 | +3.0 | +9.6 | -16.3 | +5.7 | +12.3 | +5.7 | +7.4 |
| **2-shot** | | | | | | | | | | | | |
| CLIP+KNN | 36.1 | 82.9 | 11.7 | 44.6 | 58.7 | 41.2 | 58.5 | **78.9** | 40.9 | 54.1 | 49.0 | 50.6 |
| LLaVA1.5 Finetuning | - | 53.1 | 24.9 | - | 48.2 | 22.3 | 38.7 | 10.03 | - | 38.2 | 16.3 | - |
| RAR (LLaVA1.5) | **46.8** | **89.2** | **27.9** | **53.1** | **68.6** | **47.9** | **66.5** | 54.7 | **45.9** | **65.4** | **54.7** | **57.4** |
| Δ | +10.7 | +6.3 | +16.2 | +8.5 | +9.9 | +6.7 | +8.0 | -24.2 | +5.0 | +11.3 | +5.7 | +6.8 |
| **4-shot** | | | | | | | | | | | | |
| CLIP+KNN | 42.1 | 87.9 | 14.2 | 51.4 | 67.6 | 47.5 | 64.6 | **84.5** | 49.2 | 62.6 | 55.6 | 57.0 |
| LLaVA1.5 Finetuning | - | 88.4 | 24.9 | - | 48.2 | 46.6 | 58.9 | 13.2 | - | 66.4 | 28.9 | - |
| RAR (LLaVA1.5) | **51.0** | **92.1** | **27.7** | **58.8** | **74.8** | **53.9** | **69.6** | 80.4 | **54.4** | **71.4** | **60.9** | **63.2** |
| Δ | +9.9 | +4.2 | +13.5 | +7.4 | +7.2 | +6.4 | +5.0 | -4.1 | +5.2 | +8.8 | +5.3 | +6.2 |
| **8-shot** | | | | | | | | | | | | |
| CLIP+KNN | 47.6 | 90.6 | 28.2 | 56.8 | 72.8 | 53.2 | 68.3 | **89.5** | 56.1 | 68.3 | 61.8 | 63.0 |
| LLaVA1.5 Finetuning | - | 92.1 | 24.9 | - | 48.2 | 54.7 | 66.5 | 30.1 | - | 72.5 | 46.1 | - |
| RAR (LLaVA1.5) | **56.5** | **93.5** | **46.9** | **63.4** | **81.5** | **59.3** | **74.3** | 87.3 | **61.2** | **76.6** | **67.7** | **69.8** |
| Δ | +8.9 | +2.9 | +18.7 | +6.6 | +8.7 | +6.1 | +6.0 | -2.2 | +5.1 | +8.3 | +5.9 | +6.8 |
| **16-shot** | | | | | | | | | | | | |
| CLIP+KNN | 52.0 | 92.4 | 35.0 | 61.2 | 78.7 | 57.5 | 70.6 | 92.1 | 63.2 | 71.8 | 68.3 | 67.5 |
| LLaVA1.5 Finetuning | - | 94.1 | 24.9 | - | 50.6 | 63 | 74.7 | 59.0 | - | - | 62.4 | - |
| RAR (LLaVA1.5) | **60.3** | **94.1** | **53.1** | **68.0** | **84.8** | **63.7** | **75.9** | **92.1** | **67.8** | **79.4** | **72.7** | **73.8** |
| Δ | +8.3 | +1.7 | +18.1 | +6.8 | +6.1 | +6.2 | +5.3 | +0.0 | +4.6 | +7.6 | +4.4 | +6.3 |

sification tests from 1-shot to 16-shot, tests for top-5 accuracy, and we have further expanded our memory to explore the potential capabilities of RAR.

**More Discussion about Motivation.** In the field of image classification, especially when facing the challenges of fine-grained image categorization, can MLLMs prove competent and effective? To further explore the potential of MLLMs in image classification tasks, we employed the GPT-4V

Table 8: Evaluation on 11 datasets, reporting the top-5 accuracy. We use the 4-shot setting.

| Method | Common | | | | | | | Fine-Grained | | | | |
|---|---|---|---|---|---|---|---|---|---|---|---|---|
| | ImageNet | Caltech101 | RAF-DB | SUN397 | EuroSAT | DTD | UCF-101 | Flower102 | StanfordCars | Food101 | OxfordPets | Average |
| CLIP+KNN | 67.1 | 97.6 | 48.0 | 78.9 | 91.5 | 70.5 | 85.4 | 96.5 | 79.1 | 86.2 | 87.6 | 80.8 |
| RAR (LLaVA1.5) | **69.7** | **97.7** | **53.8** | **80.1** | **92.5** | **71.9** | **86.2** | **96.5** | **79.1** | **87.7** | **88.1** | **82.1** |
| Δ | +2.6 | +0.1 | +5.8 | +1.2 | +1.0 | +1.4 | +0.8 | +0.0 | +0.0 | +1.5 | +0.5 | +1.3 |

Table 9: Evaluation on 11 datasets, reporting the top-1 accuracy. The GPT4V (OpenAI, 2023) results are copied from (Wu et al., 2023).

| Method | Common | | | | | | | Fine-Grained | | | | |
|---|---|---|---|---|---|---|---|---|---|---|---|---|
| | ImageNet | Caltech101 | RAF-DB | SUN397 | EuroSAT | DTD | UCF-101 | Flower102 | StanfordCars | Food101 | OxfordPets | Average |
| GPT-4V | 62.0 | 95.5 | 58.5 | 57.7 | 36.2 | 59.1 | **81.6** | 70.6 | 58.3 | 80.1 | **92.6** | 68.4 |
| RAR (LLaVA1.5) | 73.4 | 94.6 | **73.8** | 70.6 | **93.3** | 71.9 | 79.1 | 95.6 | 72.6 | 86.2 | 79.9 | 81.0 |
| Δ | +11.4 | -0.9 | +15.3 | +12.9 | +57.1 | +12.8 | -2.5 | +25.0 | +14.3 | +6.1 | -12.7 | +12.6 |
| RAR (Intern-IXC2) | 71.5 | 94.4 | 72.7 | 69.7 | 91.7 | 69.9 | 77.6 | 93.2 | 65.4 | 83.9 | 79.3 | 79.0 |
| Δ | +9.5 | -1.1 | +14.2 | +12.0 | +55.5 | +10.8 | -4.0 | +22.6 | +7.1 | +3.8 | -13.3 | +10.6 |
| RAR (Qwen-VL) | **75.8** | **95.5** | 66.0 | **72.7** | 90.7 | **72.5** | 81.4 | **97.5** | **81.6** | **87.2** | 88.1 | **82.6** |
| Δ | +13.8 | +0.0 | +7.5 | +5.0 | +54.5 | +13.4 | -0.2 | +26.9 | +23.3 | +7.1 | -4.5 | +14.2 |

model to test selected images from our fine-grained datasets. Initially, we used the CLIP+KNN method to select 5 candidate images and their categories for a single image, ensuring that these candidates are at the top-5 in similarity among all images in memory, thus guaranteeing minimal differences between the chosen categories. Additionally, we intentionally selected examples that CLIP failed to classify correctly, increasing the complexity of the task. Subsequently, we presented these images and categories to GPT-4V, utilizing the prompt described in Sec. B, prompting GPT-4V to rank all categories by similarity. During this process, we also requested GPT-4V to provide the rationale for its classifications, allowing us to analyze the specific role of MLLMs in classification tasks based on the reasons provided by GPT-4V. Fig. 13 and Fig. 8 presents several examples of five fine-grained classification datasets.

From the examples in Fig. 13 and Fig. 8, it is evident that GPT-4V is capable of effectively analyzing the main feature information of objects in images during fine-grained image classification tasks. For instance, it identifies key characteristics such as "**coupe**" (a two-door car), "**long fuselage**" (long body of an aircraft), and "**prominent ears**" (noticeably protruding ears), which are crucial for distinguishing between similar categories. Sometimes, these detailed aspects may be overlooked by the CLIP model, leading to classification errors. Therefore, adopting a method of initial retrieval followed by deeper analysis, firstly filtering through the numerous fine-grained categories and then using MLLMs for further examination to select the most accurate answer, proves to be an effective approach for fine-grained image classification tasks.

Simultaneously, we assessed CLIP's accuracy in handling a variety of classification datasets. We selected six datasets: Caltech101, Flower102, RAF-DB, Pets37, DTD, and UCF101, and tested the CLIP+KNN method for top 1, 5, 10, 15, and 20 accuracy, with results presented in Fig. 9. We observed that as the top-k value increased, the classification accuracy improved rapidly, reaching over 90% in four of the six datasets when top-k reached 10. This indicates that CLIP shows significant advantages as the number of predicted categories increases, complementing MLLMs' ability to discern among similar categories.

Table 10: Evaluation on 11 datasets, reporting the top-1 accuracy. We use the CLIP ViT-L/14@336 as feature extractor and RAR is based on LLaVA 1.5.

| Method | Common | | | | | | | Fine-Grained | | | | |
|---|---|---|---|---|---|---|---|---|---|---|---|---|
| | ImageNet | Caltech101 | RAF-DB | SUN397 | EuroSAT | DTD | UCF-101 | Flower102 | StanfordCars | Food101 | OxfordPets | Average |
| **4-shot** | | | | | | | | | | | | |
| CLIP+KNN | 52.2 | 92.4 | 24.7 | 56.2 | 68.3 | 52.5 | 72.6 | **92.3** | 62.4 | 74.1 | 67.0 | 65.0 |
| RAR (LLaVA1.5) | **58.4** | **93.6** | **46.3** | **61.8** | **73.5** | **58.5** | **75.5** | 83.2 | **70.8** | **79.0** | **68.4** | **69.9** |
| Δ | +6.2 | +1.2 | +21.6 | +5.6 | +5.2 | +6.0 | +2.9 | -9.1 | +8.4 | +4.9 | +1.4 | +4.9 |
| **8-shot** | | | | | | | | | | | | |
| CLIP+KNN | 57.8 | 94.4 | 41.0 | 61.3 | 78.9 | 57.0 | 76.2 | **95.8** | 63.1 | 80.2 | 73.1 | 70.8 |
| RAR (LLaVA1.5) | **63.2** | **95.0** | **57.6** | **66.9** | **84.3** | **62.8** | **79.1** | 93.2 | **70.8** | **83.5** | **73.7** | **75.5** |
| Δ | +5.4 | +0.6 | +16.6 | +5.6 | +5.4 | +5.8 | +2.9 | -2.6 | +7.7 | +3.3 | +0.6 | +4.7 |

| | $k = 3$ | $k = 4$ | $k = 5$ | $k = 6$ | $k = 7$ |
|---|---|---|---|---|---|
| DTD | 70.27 | 71.34 | 71.93 | 71.93 | **71.99** |
| Flowers102 | **96.18** | 95.57 | 95.62 | 95.66 | 95.57 |
| Oxford-pets | 80.21 | **80.38** | 79.91 | 79.72 | 79.42 |
| Eurosat | 92.38 | 92.48 | **93.28** | 92.52 | 92.59 |
| Average | 84.76 | 84.96 | **85.19** | 84.96 | 84.90 |

Table 11: **Ablation studies** about the selection of the hyper-parameter $k$.

Following the experimental design in Fig. 9, we used MLLMs to rank categories when expanding the number of categories. We chose two datasets, Caltech101 and Flowers102, and used 10, 15, 20, 25, 30 categories as input to MLLMs, ensuring these included the correct category. As shown in Fig. 10, the distinction ability of MLLMs gradually decreased as the number of categories input into MLLMs increased.

Hence, we found that MLLMs and CLIP have complementary advantages in classification tasks. CLIP initially narrows down the correct answer to a smaller set through preliminary screening, while MLLMs can finely select the correct answer from this set. Our RAR combines the strengths of both CLIP and MLLMs, first finding likely correct candidates through CLIP and retrieval, and then accurately selecting the correct answer through MLLMs' ranking, thus achieving outstanding results across multiple classification datasets.

**More Evaluation Results.** In our few-shot image classification experiments, we employed the CLIP B/16 model to extract embeddings from n images in each category, which were then stored in memory for testing the accuracy of n-shot experiments. To accelerate retrieval speed, we initially use the HNSW algorithm to transform the original 576-dimensional vectors into 64-dimensional indices before storing the image embeddings in memory. HNSW is a commonly used Approximate Nearest Neighbor (ANN) algorithm, primarily aimed at quickly finding the k nearest elements to a query in a large set of candidates. To demonstrate the effectiveness of our method, we included results from 1-shot, 2-shot, and 16-shot experiments in the supplementary materials, alongside the results of 4-shot and 8-shot experiments, all of which are presented in Tab. 7.

From the 1-shot to 16-shot experiments, RAR's results showed an improvement over the CLIP+KNN method by **7.4%**, **6.8%**, **6.2%**, **6.8%**, and **6.3%** respectively, averaging a **6.7%** percentage point increase, and significantly outperforming the performance of the LLaVa model itself. This outcome demonstrates the excellence of RAR in image classification tasks (including fine-grained image classification), achieved by integrating the strengths of MLLMs and retrieval techniques.

**Top-5 Accuracy Results.** Moreover, in the experiments conducted for our paper, we selected the top 5 retrieved results for ranking. To test the scalability of this method, we conducted a new experiment using the top 10 retrieved results, ranking these ten categories and then assessing the accuracy of the top 5. In this experiment, we utilized a 4-shot setting, the result is shown in Tab. 8.

Table 12: **Cropping ablation of CLIP** (Radford et al., 2021) **zero-shot classification** on LVIS (Gupta et al., 2019) with ground truth proposals. Different behaviors can be seen before and after blurring with respect to different object scales.

| Crop scale | Blurring | 1.0 | 1.2 | 1.4 | 1.6 | 1.8 | 2.0 | 2.2 | 2.4 | 2.6 | 2.8 | 3.0 |
|---|---|---|---|---|---|---|---|---|---|---|---|---|
| AP | ✗ | 46.7 | **47.0** | 46.6 | 46.4 | 43.4 | 43.0 | 40.9 | 40.7 | 37.7 | 37.1 | 36.2 |
| | ✓ | 47.9 | 51.3 | 52.2 | **53.9** | 53.3 | 52.9 | 52.6 | 51.8 | 51.2 | 50.3 | 49.8 |
| $AP_s$ | ✗ | 39.5 | 40.9 | 44.6 | **44.8** | 44.4 | 44.2 | 42.9 | 43.3 | 41.2 | 40.5 | 39.8 |
| | ✓ | 33.6 | 35.2 | 41.4 | 43.2 | 45.6 | 46.3 | 46.7 | 46.9 | **47.4** | 47.4 | 47.3 |
| $AP_m$ | ✗ | **61.5** | 61.3 | 56.4 | 55.2 | 49.5 | 48.6 | 44.4 | 43.7 | 39.9 | 39.0 | 38.5 |
| | ✓ | 63.5 | 64.2 | 66.1 | **68.3** | 65.2 | 64.2 | 63.4 | 62.2 | 61.0 | 59.2 | 58.6 |
| $AP_l$ | ✗ | **59.1** | 57.2 | 51.1 | 50.1 | 45.6 | 44.4 | 41.4 | 40.9 | 38.0 | 37.8 | 37.2 |
| | ✓ | **72.4** | 71.3 | 69.5 | 69.6 | 67.0 | 65.2 | 62.9 | 60.7 | 59.6 | 57.4 | 55.2 |

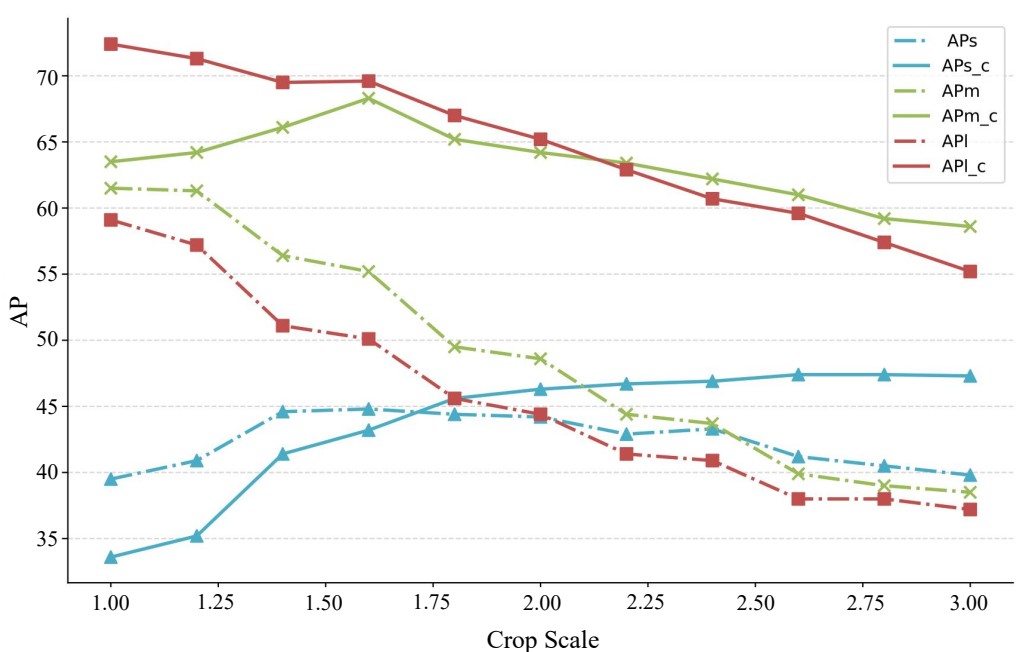

Figure 11: **Metric curve visualization of CLIP** (Radford et al., 2021) **zero-shot classification** on LVIS (Gupta et al., 2019) with ground truth proposals. Different behaviors can be seen before and after blurring with respect to different object's scales.

The final results demonstrate that although the top 5 accuracy achieved by CLIP+KNN was already high, our RAR method still managed to make comprehensive improvements on this basis. The average top 5 accuracy across eleven datasets increased by **1.3**%.

**Extension to the whole Training Set.** To further explore the potential of RAR, we expanded the memory size to include all images from the training set stored in memory. We then compared the performance of RAR under this setup with that of GPT-4V across multiple image classification datasets. The results are presented in Tab. 9.

The results in Tab. 9 show that, regardless of whether the base model is LLaVa, Intern-IXC2, or Qwen-VL, RAR significantly outperforms GPT-4V in terms of accuracy. Across eleven datasets, the average precision of RAR exceeds that of GPT-4V by 12.5 percentage points. It is observed that even 7B MLLMs, when integrated into the RAR pipeline, far surpass the classification capabilities of GPT-4V across multiple image classification datasets.

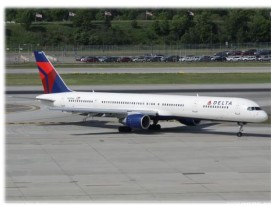

**Question:** Please play the role of a classification expert, suppose you are given a picture and 5 categories, then you need to sort the provided 5 categories from high to low according to the top 5 similarity with the input image.
For example, you are given a picture of a chair, and the provided categories are ['armchair', 'recliner', 'chair', 'rocking_chair', 'loveseat'], the sorted categories should be like: "['chair', 'armchair', 'rocking_chair', 'recliner', 'loveseat']", and your answer is: "['chair', 'armchair', 'rocking_chair', 'recliner', 'loveseat']", no more information.
Now, according to the input pictures, sort the provided categories: ['767-400', '747-400', '767-400', '757-300', 'A319'].

Input Category: 757-300

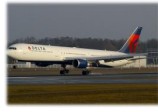 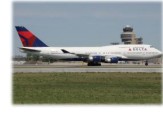 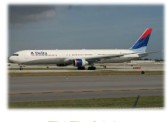 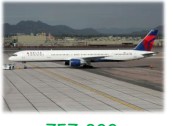 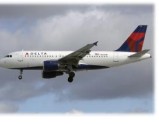

767-400      747-400      767-400      757-300      A319

**Answer:** ['757-300', '767-400', '747-400', '767-400', 'A319']

--------------------------------------------------------------------------------

**Question:** The above is an example of a ranking. *Please explain the reason for your ranking results(Answer).*

**Answer:** In the given image, there is a large commercial airplane parked on the runway. The image is described as a Delta airplane, which is a specific type of aircraft. The categories provided are related to different types of airplanes, including the Delta airplane.

To sort the categories from high to low according to the top 5 similarity with the input image, we need to consider the visual features of the airplane and the categories. In this case, the Delta airplane is a large commercial airplane, which is a specific type of aircraft. The categories related to other types of airplanes, such as the 767-400, 747-400, 757-300, and A319, are not as similar to the Delta airplane as the categories related to other Delta airplanes, such as the 767-400 and 757-300. Therefore, the categories related to other types of airplanes should be ranked lower than the categories related to the Delta airplane.

The sorted categories are: ['757-300', '767-400', '747-400', '767-400', 'A319'], which reflects the similarity between the input image and the categories related to the Delta airplane.

Figure 12: **Interpretability.** Our model provides interpretability for the ranking process.

**More Feature Extractor results.** In the few-shot image classification experiment, we primarily used the CLIP-B/16. To verify the generalization capability of the method, we conducted experiments using the more powerful CLIP ViT-L/14@336 as the feature extractor. The results, as shown in Tab. 10, demonstrate that RAR still consistently outperforms the baseline.

**Ablation Study on Effects of the Parameter $k$.** We delve into the impact of the hyper-parameter $k$ on few-shot image recognition setting, as detailed in Tab. 13. We report the results of RAR with the LLaVA1.5 as the MLLM. Our findings reveal that our RAR demonstrates a remarkable robustness to variations in $k$, with only minor differences observed across a broad spectrum of values from 3 to 7. Such a consistency suggests that RAR's ability to generalize from a few examples is not significantly influenced by the choice of $k$. Consequently, based on the averaged results, we select $k = 5$ as the default choice.

## C.3 ZERO-SHOT REGION RECOGNITION

We carefully study how to adapt CLIP and MLLMs pretrained on full images to region-level recognition tasks. Zero-shot LVIS (Gupta et al., 2019) AP metric under different crop scales and object scales are reported in Fig. 11 and Tab. 12. Based on this experiment, we conclude with two major observations: Firstly, a proper amount of blurring can significantly improve classification accuracy. This trick can help leave enough context information while keeping the foreground object prominent. Secondly, for objects with different scales, different crop scales should be adapted to maximize classification accuracy. As shown in Fig. 11, after blurring, Different object scale AP curves behave differently with respect to crop scale. We contribute this phenomenon to the resolution shift of CLIP input images. Therefore, we make two adaptations for CLIP and MLLMs for region-level recognition: Gaussian blurring and adaptive crop scale. We adopt the hyperparameters of these two tricks on the LVIS training set and find these adaptions not only fit for the LVIS validation set but also other detection datasets like V3Det (Wang et al., 2023a).

|  | DTD | Flowers102 | Oxford-pets | Eurosat | Average |
|---|---|---|---|---|---|
| LLaVa baseline | 46.6 | 13.2 | 28.9 | 48.2 | 34.2 |
| CLIP+KNN baseline | 68.4 | 95.5 | 75.6 | 90.5 | 82.5 |
| RAR ($k = 3$) | 70.3 | 96.2 | 80.2 | 92.4 | 84.8 |
| RAR ($k = 4$) | 71.3 | 95.6 | **80.4** | 92.5 | 85.0 |
| RAR ($k = 5$) | 71.9 | 95.6 | 79.9 | **93.3** | **85.2** |
| RAR ($k = 6$) | 71.9 | **95.7** | 79.7 | 92.5 | 85.0 |
| RAR ($k = 7$) | **72.0** | 95.6 | 79.4 | 92.6 | 84.9 |
| Upper bound | 89.0 | 99.2 | 97.0 | 98.3 | 95.8 |

Table 13: **Limitation Study.** We present the baselines of LLaVa and CLIP, as well as the upper limit of the model and RAR's results under different k values.

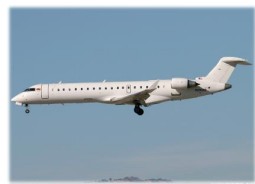

**Question:** Please play the role of a classification expert, suppose you are given a picture and 5 categories, then you need to sort the provided 5 categories from high to low according to the top 5 similarity with the input image.
For example, you are given a picture of a chair, and the provided categories are ['armchair', 'recliner', 'chair', 'rocking_chair', 'loveseat'], the sorted categories should be like: "['chair', 'armchair', 'rocking_chair', 'recliner', 'loveseat']", and your answer is: "['chair', 'armchair', 'rocking_chair', 'recliner', 'loveseat']", no more information.
Now, according to the input pictures, sort the provided categories: ['Global Express', 'ERJ 145', 'ERJ 135', 'Global Express', 'Falcon 2000'].

Input Category: CRJ-700

Retrieved Category:

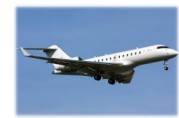 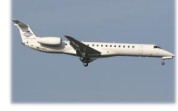 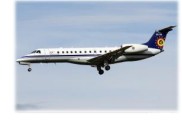 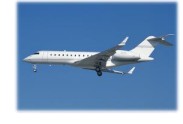 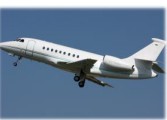

Global Express     ERJ 145     ERJ 135     Global Express     Falcon 2000

The differences in fine-grained categories are so subtle that even the results retrieved by CLIP are all incorrect.

- - - - - - - - - - - - - - - - - - - - - - - - - - - - - - - - - - - - - - - - - -

**Question:** Please play the role of a classification expert, suppose you are given a picture and 5 categories, then you need to sort the provided 5 categories from high to low according to the top 5 similarity with the input image.
For example, you are given a picture of a chair, and the provided categories are ['armchair', 'recliner', 'chair', 'rocking_chair', 'loveseat'], the sorted categories should be like: "['chair', 'armchair', 'rocking_chair', 'recliner', 'loveseat']", and your answer is: "['chair', 'armchair', 'rocking_chair', 'recliner', 'loveseat']", no more information.
Now, according to the input pictures, sort the provided categories: ['EMB-120', 'DHC-8-300', 'Embraer Legacy 600', 'DHC-8-100', 'Cessna 525'].

Input Category: DHC-8-100

Retrieved Category:

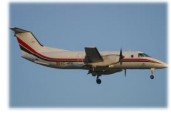 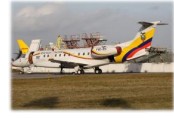 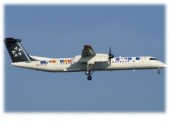 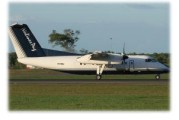 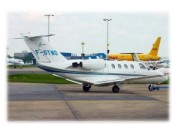

EMB-120     DHC-8-300     Embraer Legacy 600     DHC-8-100     Cessna 525

**Answer:** ['DHC-8-300', 'DHC-8-100', 'EMB-120', 'Embraer Legacy 600', 'Cessna 525']

MLLM tends to make errors when encountering categories beyond its knowledge scope.

Figure 13: **Error Analysis.** RAR is prone to errors in the following two scenarios: 1. The differences in fine-grained categories are so subtle that even the results retrieved by CLIP are all incorrect. 2. MLLM tends to make errors when encountering categories beyond its knowledge scope.

