# OpenReview forum: "RAR: Retrieving And Ranking Augmented MLLMs for Visual Recognition"
_ICLR.cc/2025/Conference — Submitted to ICLR 2025_

### Official Review · Reviewer_nzRY · 2024-10-29

**Soundness:** 4
**Presentation:** 4
**Contribution:** 2
**Rating:** 6
**Confidence:** 4

**Summary:**

This paper introduces RAR (Retrieving and Ranking), a method to enhance multimodal large language models (MLLMs) in visual recognition tasks. RAR combines CLIP’s broad retrieval ability with MLLMs' fine-grained differentiation abilities. It retrieves candidate categories from external memory based on the input image, which the MLLM then ranks to make a final prediction. Extensive experiments show that RAR significantly improves performance across various visual benchmarks, including fine-grained classification, few-shot recognition, and zero-shot object detection.

**Strengths:**

- Well-written and straightforward
- Easy to understand with a clear methodology
- Achieves strong results on the proposed benchmark
- Thoroughly tested across multiple visual benchmarks, including fine-grained classification, few-shot recognition, and zero-shot object detection

**Weaknesses:**

- Limited novelty in the proposed method.

**Questions:**

- Could you clarify why vanilla LLaVa results are not included in Table 1? Additionally, for zero-shot image classification, is there a comparison to vanilla CLIP using only category names?

---

> ### Author Response · Authors · 2024-11-23
> **Authors Rebuttal to Reviewer nzRY**
>
> Dear Reviewer nzRY,
>
> We sincerely thank you for your thorough feedback: the paper is well-written, easy to understand with a clear methodology and achieves strong results on the proposed benchmark. We have incorporated all your feedback in our revised manuscript. **All new material added to the revised manuscript has been highlighted in red text for better visibility.**
>
> > Q1: Could you clarify why vanilla LLaVa results are not included in Table 1?
>
> A1: LLaVA is not designed for zero-shot fine-grained visual recognition, which requires classifying images into many specific categories. Directly using LLaVA for such tasks would yield significantly inferior performance compared to previous baselines (e.g., CaSED, FineR). Therefore, we excluded LLaVA from Table 1 to maintain a fair comparison.
>
> >  Q2: Additionally, for zero-shot image classification, is there a comparison to vanilla CLIP using only category names?
>
> A2：Thanks for your suggestion. We've included the zero-shot performance of vanilla CLIP using only category names below. While Table 2 primarily focuses on few-shot image classification, where a few sample images are provided, zero-shot classification relies solely on category names. Comparing the results, We found that on some fine-grained datasets (e.g., Eurosat, DTD, UCF101, Flowers102), our method significantly outperforms CLIP+category. This highlights the effectiveness of our approach, particularly when limited training data is available.
>
> | Dataset     | CLIP+category | RAR   |
> |-------------|---------------|-------|
> | Eurosat     | 47.7          | 74.8  |
> | DTD         | 43.1          | 53.9  |
> | UCF101      | 67.5          | 69.6  |
> | Flower102   | 70.7          | 80.4  |

---

> ### Author Response · Authors · 2024-11-26
>
> Dear Reviewer nzRY,
>
> Thank you again for your time and valuable feedback on our paper. We have carefully addressed all your comments in our rebuttal and would greatly appreciate it if you could review our response and consider adjusting your score accordingly.
>
> Your support means a lot to us, and we are grateful for your consideration.
>
> Best regards,
>
> Authors

---

> > ### Comment · Reviewer_nzRY · 2024-11-26
> >
> > The authors have thoroughly addressed all of my concerns and questions from the initial review. I am maintaining my original score and increasing its confidence level to 4, reflecting my strengthened belief in the quality and significance of the work.

---

> > > ### Author Response · Authors · 2024-11-26
> > >
> > > Dear Reviewer nzRY,
> > >
> > > Thank you very much for your thoughtful response and for maintaining your original score 6 while increasing your confidence level to 4. We are truly grateful for your recognition of the quality and significance of our work. Your feedback and encouragement mean a great deal to us and motivate us to continue improving our research.
> > >
> > > We appreciate your time and effort in thoroughly reviewing our paper, and we are delighted that the revisions have addressed your concerns to your satisfaction.
> > >
> > > Thank you once again for your constructive feedback and for helping to make our paper stronger.
> > >
> > > Sincerely,
> > >
> > > Authors

---

### Official Review · Reviewer_FduB · 2024-11-02

**Soundness:** 2
**Presentation:** 3
**Contribution:** 1
**Rating:** 5
**Confidence:** 3

**Summary:**

The paper addresses the issues of CLIP's weak fine-grained category classification capabilities and MLLM's limitations when dealing with extensive vocabularies and fined-grained categorizations, by proposing a method named RAR. Specifically, it involves pre-storing features information about each category's images or labels in the dataset through CLIP's encoder. This stored information is then matched with the images’ features that need to be classified using either image-image kNN or image-text kNN methods to select the top-k targets. These top-k candidate targets are subsequently fed into MLLM for ranking, ultimately yielding the final prediction results.

**Strengths:**

1.The paper is well-written, and the method's details are well-explained. The drawings are appealing and intuitive.
2.The experiments are comprehensive and conducted on a sufficient number of datasets, and improvements have been achieved on many datasets.
3.The paper includes a complete evaluation and ablation study to understand the impact of the propose components.

**Weaknesses:**

The method does not fundamentally address the fine-grained categorization and extensive vocabulary classification issues of CLIP and MLLM. Essentially, it is more like a simple combination of the two, which lacks novelty and not suitable for ICLR.

1. Although the paper’s title is about augmenting MLLM, it is mostly just a simple application of MLLM. The SFT and in-context learning techniques used in the thesis only improve the MLLM's ability to maintain output formats, but do not actually enhance its classification ability. This is why the SFT effects on different datasets in Table 5 are not significantly different.

2. The motivation was the improvement in fine-grained categories and vast vocabulary classification for CLIP and MLLM, but since no substantial optimization was done for CLIP and MLLM, and their classification abilities for this scenario are relatively weak, so the improvement of their combination is not significant, as shown in Table 1 against FineR and Table 5. The reason why the improvement was significant in Table 4 was because the object categories in LVIS were 1,000, which was not enough, and could not effectively prove the motivation.

3. The performance increase in Table 2 is all compared with CLIP+KNN, which makes people feel that RAR improves CLIP rather than MLLM, and the performance increase here is suggested to be worse than the sub-optimal result.

**Questions:**

When the K of K-NN is determined, the upper limit of the method's capability is determined by CLIP (whether the prediction result is in topk), and MLLM only helps the method approach this upper limit. In some cases, even the performance of CLIP will limit the play of MLLM (if the correct category is not in topk). Also, poor MLLM performance can also undermine CLIP's capabilities (see Table 2 Flower102 dataset).

Therefore, I want to see the ablation experiments under different k settings. In-context learning can be used if SFT is too time-consuming.

---

> ### Author Response · Authors · 2024-11-23
> **Authors Rebuttal to Reviewer FduB（1/2）**
>
> Dear Reviewer FduB,
>
> We sincerely thank you for your thorough feedback: the paper is well-written and the method's details are well-explained. We have incorporated all your feedback in our revised manuscript. **All new material added to the revised manuscript has been highlighted in red text for better visibility.**
>
> > Q1: Although the paper’s title is about augmenting MLLM, it is mostly just a simple application of MLLM. The SFT and in-context learning techniques used in the thesis only improve the MLLM's ability to maintain output formats, but do not actually enhance its classification ability. This is why the SFT effects on different datasets in Table 5 are not significantly different.
>
> A1: While our paper explores the synergy between Vision-Language Models (VLMs) and Multi-Modal Language Models (MLLMs) for visual recognition, our primary contribution lies in using MLLMs for **re-ranking** retrieved visual examples. We do not rely on SFT to enhance classification performance. The SFT and in-context learning techniques we employ are specifically designed to enhance MLLM's ability to maintain consistent output formats during the re-ranking process. This ensures that the final predictions align with the desired task and dataset.
>
> As demonstrated in Table 5, in-context learning, which does not involve SFT, proves effective in improving re-ranking performance. This highlights the power of MLLMs in adapting to new tasks without extensive fine-tuning. While SFT can further refine the model's performance, our results indicate that in-context learning is a strong baseline that can be leveraged for a variety of visual recognition tasks.
>
> > Q2: The motivation was the improvement in fine-grained categories and vast vocabulary classification for CLIP and MLLM, but since no substantial optimization was done for CLIP and MLLM, and their classification abilities for this scenario are relatively weak, so the improvement of their combination is not significant, as shown in Table 1 against FineR and Table 5. The reason why the improvement was significant in Table 4 was because the object categories in LVIS were 1,000, which was not enough, and could not effectively prove the motivation.
>
> A2: We combined the strengths of CLIP and MLLMs in handling large vocabularies and fine-grained categories. Experiments were conducted in three directions: fine-grained visual recognition tasks, few-shot image classification tasks, and zero-shot object detection tasks. The results demonstrate that our method achieves significant performance improvements across all tasks.
> In Table 3 and Table 4, we present the results of the LVIS and V3Det object detection datasets. The former contains over 1,000 categories, while the latter includes over **13,000 categories**. We believe that such a vast vocabulary is sufficient to substantiate our claims.
>
> > Q3: The performance increase in Table 2 is all compared with CLIP+KNN, which makes people feel that RAR improves CLIP rather than MLLM, and the performance increase here is suggested to be worse than the sub-optimal result.
>
> A3:
> We did not merely compare against the CLIP baseline. In Table 2, CLIP+KNN and LLaVA 1.5+finetuning represent the baselines for CLIP and LLaVA, respectively. The LLaVA model performs significantly worse than CLIP when dealing with classification datasets containing a large number of categories. However, RAR achieves substantial performance improvements over both baselines. This reflects our motivation to combine the strengths of CLIP and MLLMs, enabling better results in visual perception tasks by using the advantages of both models.

---

> ### Author Response · Authors · 2024-11-23
> **Authors Rebuttal to Reviewer FduB（2/2）**
>
> > Q4: When the K of K-NN is determined, the upper limit of the method's capability is determined by CLIP (whether the prediction result is in topk), and MLLM only helps the method approach this upper limit. In some cases, even the performance of CLIP will limit the play of MLLM (if the correct category is not in topk). Also, poor MLLM performance can also undermine CLIP's capabilities (see Table 2 Flower102 dataset). Therefore, I want to see the ablation experiments under different k settings. In-context learning can be used if SFT is too time-consuming.
>
> A4: You've raised an important point. The choice of K in K-NN significantly impacts RAR's performance, as it determines the upper limit of the method's potential.
>
> To investigate this further, we've conducted ablation studies in Table 13 with varying K values (K = 3, 4, 5, 6, 7) and compared the results to the following baselines:
>
> - Upper-bound: The CLIP top-7 accuracy, represents the ideal scenario where all retrieved results are ranked correctly.
>
> - RAR with varying K: The top-a accuracy of RAR for different K values.
>
> - CLIP (VLM only): The top-1 accuracy of CLIP, showcases the performance of the VLM alone.
>
> - LLaVA (MLLM only): The top-1 accuracy of LLaVA, demonstrates the performance of the MLLM alone.
>
> Our analysis reveals the following insights:
>
> (1) **Small K values (K < 5)**: When K is small, the correct category may not be included in the top-K retrieved results by CLIP. In such cases, RAR's performance gain over CLIP is limited, as MLLM's re-ranking capabilities are constrained by the initial retrieval.
>
> (2) **Large K values**: As K increases, the challenge for MLLM becomes more significant. Re-ranking a large number of candidates accurately requires strong language understanding and reasoning abilities, which may exceed the current capabilities of MLLMs.
> In future work, exploring more advanced retrieval techniques or enhancing MLLM's capabilities could further improve RAR's performance, especially in scenarios with large K values.

---

> ### Author Response · Authors · 2024-11-26
>
> Dear Reviewer FduB,
>
> Thank you again for your time and valuable feedback on our paper. We have carefully addressed all your comments in our rebuttal and would greatly appreciate it if you could review our response and consider adjusting your score accordingly.
>
> Your support means a lot to us, and we are grateful for your consideration.
>
> Best regards,
>
> Authors

---

> ### Author Response · Authors · 2024-11-27
>
> Dear Reviewer FduB,
>
> I hope this email finds you well. I’m writing to kindly follow up on the review of our paper.
>
> We are encouraged to share that one of the reviewers, nzRY, has maintained his original score of 6 and increased the confidence level to 4, reflecting his strengthened belief in the quality and significance of our work.
>
> We sincerely value your feedback and would greatly appreciate it if you could share your comments and scoring at your earliest convenience. Your insights are critical for the timely progress of the review process.
>
> Thank you very much for your time and support.
>
> Best regards,
>
> Authors

---

> > ### Comment · Reviewer_FduB · 2024-11-28
> >
> > The author's reply solves part of my problem.
> > However, as I questioned and author’s reply, SFT and context learning only optimize the format of MLLM output, and do not improve its ability to classify. So I share the same view as reviewer 5xbU, since MLLM is only used for re-ranking final predictions, and the contribution of MLLM is too small in the whole framework, which also lacks novelty.
> > In my opinion, a big room for improvement is in Fast Retrieval Optimization, and if the author don't just use the HNSW methodology but design his own Retrieval module and the effect can surpasses HNSW, then the contribution will be greatly increased. At present, it seems that the paper is more like a combination of CLIP and MLLM using HNSW, and the innovation is still not enough.

---

> > > ### Author Response · Authors · 2024-11-28
> > >
> > > Dear Reviewer FduB,
> > >
> > > We are truly delighted to hear that our revisions have addressed part of your concerns, and we are incredibly grateful that you have increased your score.
> > >
> > > Here we attempt to address the remaining concerns in your response. While we understand your point regarding the contribution of MLLM, we believe there are important aspects that we need to clarify further to demonstrate the novelty and significance of our approach.
> > >
> > > > Q5-1: ..., as I questioned and author’s reply, SFT and context learning only optimize the format of MLLM output, and do not improve its ability to classify.
> > >
> > > A5-1: If we treat the MLLM and our multi-modal retriever as an entire system (please do not look at MLLM in isolation), we *greatly improve the ability* of the whole system to diverse tasks (*classification* and *detection*) and diverse datasets, which is challenging for using the MLLM alone or CLIP alone.
> > >
> > > `SFT and context learning only optimize the format of MLLM output` is in fact an advantage of our method that *preserves the MLLM Knowledge*. By avoiding weight updates via in-context learning, we mitigate the risk of catastrophic forgetting, ensuring that the MLLM retains its original knowledge and abilities on other tasks (e.g., visual question-answering, document understanding, video understanding) and greatly improves the performance on our settings (zero-shot visual recognition and detection, etc.).
> > >
> > > We do not pursue improving the MLLM's ability to classify (which may degrade the performance of MLLM on other tasks beyond classification such as VQA). Instead, we construct a system that consists of a multi-modal retrieval and a multi-modal re-ranker for classification or detection *at a system level* without compromising the MLLM's general knowledge.
> > >
> > > > Q5-2: So I share the same view as reviewer 5xbU, since MLLM is only used for re-ranking final predictions, and the contribution of MLLM is too small in the whole framework, which also lacks novelty.
> > >
> > > A5-2: This is not true. As we discussed in A1-1, *MLLM plays a critical role in improving the performance of the whole system*. As shown in Table 2, our approach achieves a *notable performance increase of 6.2–6.8 (%)* compared to the CLIP baseline. The fine-grained reranking capability of MLLM significantly helps to address the limitations of CLIP. This level of performance can not be achieved by either CLIP or MLLM alone. This highlights the main novelty and core idea of our paper: by designing a *retrieval* and *reranking* mechanism, we use the complementary strengths of CLIP's large vocabulary classification and MLLM's fine-grained recognition, thereby achieving a visual perception capability that surpasses what either method can achieve individually.
> > >
> > > > Q6-1: In my opinion, a big room for improvement is in Fast Retrieval Optimization, and if the author don't just use the HNSW methodology but design his own Retrieval module and the effect can surpasses HNSW, then the contribution will be greatly increased.
> > >
> > > A6-1:
> > > Fast retrieval optimization is NOT our contribution. Replacing HNSW with brute-force search achieves *identical* classification accuracy or detection IoU with a slower speed. This paper is not targeted at designing fast retrieval algorithms. Instead, our primary contribution lies in
> > > - We present an in-depth analysis of the strengths and weaknesses of VLMs and MLLMs in processing fine-grained datasets, which has not been explored by previous papers (will discuss in A6-2 in detail).
> > > - We design a flexible multimodal retrieval module and an MLLM re-ranker, enabling a more comprehensive understanding of multimodal data (as discussed in A5-1).
> > >
> > > > Q6-2: At present, it seems that the paper is more like a combination of CLIP and MLLM using HNSW, and the innovation is still not enough.
> > >
> > > A6-2: This is not true, we are not `a combination of CLIP and MLLM using HNSW`. Our framework is based on an in-depth exploration of the strengths and weaknesses of CLIP and MLLM: CLIP is well-suited for large vocabulary classification and detection tasks, while MLLM excels in distinguishing fine-grained categories, thanks to its rich knowledge base. Through our cleverly designed multimodal retriever, we combine the strengths of CLIP (via retrieval) and MLLM (via reranking), using their respective advantages to achieve enhanced performance in visual perception tasks.
> > > Unlike a simple combination, our approach skillfully integrates each component into a cohesive system, overcoming VLM’s lack of fine-grained distinction and MLLM's limitations when handling large vocabulary. By combining the strengths of both,  we achieve enhanced performance that exceeds what either model can accomplish individually.

---

> > > ### Author Response · Authors · 2024-11-28
> > >
> > > > Q7: About novelty.
> > >
> > > A7: We think the reviewer mistakes in identifying the novelty of an academic paper. We recommend the authors to read [1].
> > > We believe novelty should not be limited to designing technical complex algorithms (which looks like the understanding of the reviewer FduB), because:
> > > - *A comprehensive analysis that is ignored by previous papers* can also be worth publishing. We present an in-depth exploration of the strengths and weaknesses of VLM and MLLM,, previously overlooked, directly informed the design of our multi-modal retrieval and reranker (see A6-2).
> > > - *A simple and effective approach* can also be worth publishing. Our RAR is easy to implement (as acknowledged by Reviewer nzRY, 5xbU), and can be seamlessly integrated into various MLLMs in a plug-and-play manner.
> > >
> > > [1] Novelty in Science: A guide for reviewers. Michael J. Black.

---

> ### Author Response · Authors · 2024-12-03
>
> Dear Reviewer FduB,
>
> We are sorry that we have not received a response from you since we last addressed your questions. Do you have any further concerns? We will do our utmost to address them. Given the mixed results with two ratings of 6 and one of 5, we would be grateful if you could clarify any remaining concerns that we may not have fully addressed. We truly look forward to your response. Thank you again for your time and thoughtful feedback.
>
> With sincere appreciation,
>
> The Authors

---

### Official Review · Reviewer_Zt9H · 2024-11-03

**Soundness:** 3
**Presentation:** 3
**Contribution:** 3
**Rating:** 6
**Confidence:** 2

**Summary:**

This paper introduces RAR (Retrieving And Ranking), an approach to enhance Multimodal Large Language Models (MLLMs) with a retrieving and ranking augmentation. The proposed technique aims to address the challenges faced by models like CLIP and MLLMs when applied to fine-grained visual recognition tasks and datasets with large vocabularies. By integrating a multimodal retriever and leveraging a ranking mechanism, RAR seeks to improve zero-shot and few-shot recognition accuracy across diverse datasets.

**Strengths:**

RAR’s design, particularly the use of CLIP-based retrieval augmented by MLLM ranking, is well-founded and thoughtfully justified. The use of retrieval augmentation with a multimodal memory structure effectively reduces the dependency on extensive context windows, which is a known limitation in handling large vocabularies.

**Weaknesses:**

Incorporating a detailed error analysis on retrieval failures or ranking misclassifications would provide insights into areas where RAR may need refinement, especially regarding failure cases in subtle category differentiation.

**Questions:**

The paper mentions, "Although the brute force method is inherently straightforward, its efficiency markedly diminishes as the dataset escalates to the magnitude of millions of embeddings." The HNSW method can significantly improve retrieval speed. So, how can we quantify the complexity of brute force retrieval and HNSW when the dataset scales to millions of embeddings? Please provide specific comparative results.

---

> ### Author Response · Authors · 2024-11-23
> **Authors Rebuttal to Reviewer Zt9H**
>
> Dear Reviewer Zt9H,
>
> We sincerely thank you for your thorough feedback: RAR’s design is well-founded and thoughtfully justified. We address your questions below. We have incorporated all your feedback in our revised manuscript. **All new material added to the revised manuscript has been highlighted in red text for better visibility.**
>
> > Q1：Incorporating a detailed error analysis on retrieval failures or ranking misclassifications would provide insights into areas where RAR may need refinement, especially regarding failure cases in subtle category differentiation.
>
> A1: Thanks for your suggestion. A deeper understanding of RAR's error modes can guide future improvements. As shown in Fig. 13, we've analyzed two common failure cases:
>
> - *Subtle Category Differentiation*: When fine-grained categories are visually similar, even powerful vision-language models like CLIP may struggle to retrieve accurate results. This limits RAR's ability to learn from the retrieved examples, leading to potential misclassifications.
>
> - *Knowledge Scope Limitations*: MLLM's performance can degrade when encountering categories outside its training data distribution. This can result in incorrect predictions, especially for rare or domain-specific categories.
> Despite these limitations, our experimental results demonstrate that RAR significantly outperforms state-of-the-art methods in fine-grained visual recognition, few-shot image classification, and zero-shot object recognition settings, highlighting its robustness and effectiveness.
>
> To further enhance RAR's performance, future work could explore techniques to improve the quality of retrieved examples, such as incorporating more sophisticated visual similarity measures or expanding MLLM's knowledge base.
>
> > Q2: The paper mentions, "Although the brute force method is inherently straightforward, its efficiency markedly diminishes as the dataset escalates to the magnitude of millions of embeddings." The HNSW method can significantly improve retrieval speed. So, how can we quantify the complexity of brute force retrieval and HNSW when the dataset scales to millions of embeddings? Please provide specific comparative results.
>
> A2: Assuming the total amount of data is N.
>
> (1) Brute-force Search: For two 512-dimensional vectors, calculating the dot product requires 512 multiplications and 511 additions, making the complexity of a single dot product O(512). Therefore, the overall complexity of brute-force search is O(N⋅d), where d=512. Simplifying, the complexity becomes O(N).
>
> (2) HNSW Search: In HNSW, during data insertion, neighbor connections are maintained across multiple layers of the graph, with each point connecting to log⁡N neighbors and performing distance calculations. The complexity of HNSW search mainly depends on the number of layers in the hierarchical graph and the number of edges in the graph. The approximate complexity is O(log⁡N⋅d).
> Here, log⁡N represents the process of narrowing the search range layer by layer in the hierarchical graph, and d represents the vector dimension involved in distance calculations. For 512-dimensional feature vectors, we built an index with 64 dimensions, so the query complexity becomes O(log⁡N⋅64)=O(log⁡N).
>
> While HNSW has a higher complexity for index construction, its query complexity is significantly lower than brute-force search. Especially when N is large, HNSW greatly improves retrieval efficiency.

---

> ### Author Response · Authors · 2024-11-26
>
> Dear Reviewer Zt9H,
>
> Thank you again for your time and valuable feedback on our paper. We have carefully addressed all your comments in our rebuttal and would greatly appreciate it if you could review our response and consider adjusting your score accordingly.
>
> Your support means a lot to us, and we are grateful for your consideration.
>
> Best regards,
>
> Authors

---

> ### Author Response · Authors · 2024-11-27
>
> Dear Reviewer Zt9H,
>
> I hope this email finds you well. I’m writing to kindly follow up on the review of our paper.
>
> We are encouraged to share that one of the reviewers, nzRY, has maintained his original score of 6 and increased the confidence level to 4, reflecting his strengthened belief in the quality and significance of our work.
>
> We sincerely value your feedback and would greatly appreciate it if you could share your comments and scoring at your earliest convenience. Your insights are critical for the timely progress of the review process.
>
> Thank you very much for your time and support.
>
> Best regards,
>
> Authors

---

> ### Author Response · Authors · 2024-11-28
>
> Dear Reviewer Zt9H,
>
> I hope this message finds you well. As we are in the final stages, we would greatly appreciate your valuable feedback and scoring at your earliest convenience.
>
> We sincerely understand how busy you must be, and we are deeply grateful for the time and effort you’ve already dedicated to reviewing our paper. Your support and thoughtful feedback are extremely important to us, and we would be thankful for any final comments you can provide.
>
> Thank you once again for your time and consideration.
>
> Best regards,
>
> Authors

---

> ### Author Response · Authors · 2024-12-01
>
> Dear Reviewer Zt9H.
>
> The deadline is approaching. Now two reviewers post their response to our rebuttal, and our paper received mixed results (one 6 rating and one 5 rating). We have not yet received a response from you. Could you please review our rebuttal and share your comments? Thank you.

---

> ### Author Response · Authors · 2024-12-03
>
> Dear Reviewer Zt9H.
>
> The deadline is fast approaching, and we’ve already received mixed feedback from two reviewers (one giving a rating of 6 and the other 5). However, we have not yet received your response. With only a few hours left, could you please take a moment to review our rebuttal and provide your comments? Your input would be greatly appreciated. Thank you!

---

### Official Review · Reviewer_5xbU · 2024-11-04

**Soundness:** 3
**Presentation:** 3
**Contribution:** 2
**Rating:** 5
**Confidence:** 4

**Summary:**

This paper introduces RAR framework to address limitations in fine-grained recognition by combining CLIP’s class candidates recognition with the fine-grained classification abilities of MLLMs. The authors claim that CLIP struggles with distinguishing subtle differences, while MLLMs, despite extensive pre-training, face limitations with increasing category complexity and context window size. Therefore, RAR uses a CLIP-based retriever to store memory of categories and retrieve top-k class candidates for MLLMs to rank and predict. RAR enhances MLLMs’ recognition capability, achieving improvements across fine-grained benchmarks and object detection tasks.

**Strengths:**

1.The proposed RAR framework is simple yet effective, making it easy to understand and implement.

2.The authors conducted extensive experiments. For each of their statements, motivations, and methods, they provided thorough experimental support.

3.The authors expanded their method to include object detection tasks (it can be regarded as a form of post-processing), not limited solely to fine-grained object recognition. This provides valuable insights for future research.

4.The model demonstrates improved performance on fine-grained visual recognition tasks and object detection tasks.

**Weaknesses:**

1. Relatively Limited Novelty.
The authors use off-the-shelf models, such as CLIP and LLaVA, for fine-grained vision recognition and object detection. However, “multi-query re-ranking techniques” in RAG have already been widely adopted, for example, in Re2G (Retrieve, rerank, generate) [1] and RankRAG [2]. I did not observe any specific improvements in the Retrieving or Ranking strategy tailored to the fine-grained recognition task. This limits the novelty of the proposed framework.

2. Concern of Practical Application.
Although FineR previously demonstrated the use of large models for fine-grained vision recognition, I still question the necessity of using large models. Why not use specialized/expert models with much smaller scale to accomplish this task? To my knowledge, these expert models already perform well on the tasks evaluated by the authors. Based on this concern, I believe that using MLLM for this task should allow for open-set responses (such as providing interpretability) rather than simply using MLLM for re-ranking final predictions.

3. Need more discussion of fine-tuning for re-ranking.
I noticed that the re-ranking operation requires fine-tuning the MLLM to achieve satisfactory performance. Although the authors claim that "RAR is not sensitive to changes in the fine-tuning dataset for ranking," they only conducted experiments on FGVC-Aircraft and Stanford-Cars. I believe this may be insufficient. If using a dataset with severe domain gap with the test datasets, catastrophic forgetting might occur, and I suggest that the authors discuss this issue.

[1] Glass, Michael, et al. "Re2G: Retrieve, rerank, generate." arXiv preprint arXiv:2207.06300 (2022).
[2] Yu, Yue, et al. "Rankrag: Unifying context ranking with retrieval-augmented generation in llms." arXiv preprint arXiv:2407.02485 (2024).

**Questions:**

The questions are listed in the “weakness” section.

---

> ### Author Response · Authors · 2024-11-23
> **Authors Rebuttal to Reviewer  5xbU (1/2)**
>
> Dear Reviewer 5xbU,
>
> We sincerely thank you for your thorough feedback: simple yet effective framework and extensive experiments. We address your questions below. We have incorporated all your feedback in our revised manuscript. **All new material added to the revised manuscript has been highlighted in red text for better visibility.**
>
> > Q1: Relatively Limited Novelty. The authors use off-the-shelf models, such as CLIP and LLaVA, for fine-grained vision recognition and object detection. However, “multi-query re-ranking techniques” in RAG have already been widely adopted, for example, in Re2G (Retrieve, rerank, generate) [1] and RankRAG [2]. I did not observe any specific improvements in the Retrieving or Ranking strategy tailored to the fine-grained recognition task. This limits the novelty of the proposed framework.
>
> A1: Thank you for your valuable feedback! However, our RAR is fundamentally different from RAG and should not be treated as the same thing. RAR has the following distinct differences with RAG:
>
> - (1) *Motivation*. RAG is designed to incorporate new information for text generation, whereas RAR reveals and uses the complementary strengths of VLMs' broad category coarse recognition and MLLMs' fine-grained recognition.
>
> - (2) *Framework*. RAG uses retrieved text as context prompts for LLMs, whereas RAR designs a **multi-modal retrieval** based on VLMs and further uses MLLMs for ranking the retrieved candidates.
>
> - (3) *Tasks*. RAG is for NLP, whereas RAR is designed for large-vocabulary and fine-grained visual recognition.
>
> Our contribution lies in identifying the respective strengths and weaknesses of VLMs and MLLMs in large-vocabulary and fine-grained visual recognition. Additionally, we have explored a solution that combines the two for addressing visual recognition tasks. The Re2G series and RankRAG are both highly valuable works in the RAG field, and we have added the discussion with previous RAG works in the related work section.
>
> > Q2-1: Concern of Practical Application. Although FineR previously demonstrated the use of large models for fine-grained vision recognition, I still question the necessity of using large models. Why not use specialized/expert models with much smaller scale to accomplish this task? To my knowledge, these expert models already perform well on the tasks evaluated by the authors.
>
> A2-1:
> - Our method has the advantage of **Efficiency** and **Generalization** over using specialized/expert models. Our method uses **one** versatile MLLM model to effectively address diverse benchmarks (e.g., Bird-200, ImageNet) and tasks (e.g., recognition, object detection). This unified approach offers significant advantages in terms of efficiency and generalization, as opposed to requiring specialized models tailored to specific benchmarks and tasks (e.g., FineR for Bird-200, ResNet for ImageNet, Mask R-CNN for LVIS).
> - Using MLLMs has better interpretability than using specialized/expert models (will be elaborated in A2-2).
>
> > Q2-2: Based on this concern, I believe that using MLLM for this task should allow for open-set responses (such as providing interpretability) rather than simply using MLLM for re-ranking final predictions.
>
> A2-2: We appreciate your insightful suggestion. Our method primarily employs MLLMs to refine the ranking of candidate predictions, while preserving their capacity for open-set responses and interpretability. As demonstrated in Fig. 12, MLLMs use their inherent knowledge to assist in visual perception tasks during the reranking process. This aligns seamlessly with the goal of providing explicit interpretations for model decisions.

---

> > ### Author Response · Authors · 2024-11-28
> >
> > Dear Reviewer 5xbU,
> >
> > I hope this message finds you well. As we are in the final stages, we would greatly appreciate your valuable feedback and scoring at your earliest convenience.
> >
> > We sincerely understand how busy you must be, and we are deeply grateful for the time and effort you’ve already dedicated to reviewing our paper. Your support and thoughtful feedback are extremely important to us, and we would be thankful for any final comments you can provide.
> >
> > Thank you once again for your time and consideration.
> >
> > Best regards,
> >
> > Authors

---

> ### Author Response · Authors · 2024-11-23
> **Authors Rebuttal to Reviewer 5xbU (2/2)**
>
> > Q3-1: Need more discussion of fine-tuning for re-ranking. I noticed that the re-ranking operation requires fine-tuning the MLLM to achieve satisfactory performance.
>
> A3-1: This is not true. Our method offers two complementary approaches for re-ranking: (1) in-context learning, which uses the MLLM's capabilities without altering its weights, and (2) fine-tuning, which can further improve performance through tailored training. The results in Table 5 highlight the effectiveness of in-context learning that does not require fine-tuning.
>
> > Q3-2: Although the authors claim that "RAR is not sensitive to changes in the fine-tuning dataset for ranking," they only conducted experiments on FGVC-Aircraft and Stanford-Cars. I believe this may be insufficient. If using a dataset with severe domain gap with the test datasets, catastrophic forgetting might occur, and I suggest that the authors discuss this issue.
>
> A3-2:
> - As discussed in A3-1, our method supports in-context learning for re-ranking, which eliminates the need for fine-tuning and, consequently, the risk of catastrophic forgetting. Since the MLLM weights remain unchanged, the model retains its knowledge and can effectively handle datasets with varying domain gaps.
>
> - Regarding fine-tuning, our SFT process primarily focuses on teaching the model a specific response format and enhancing its ability to follow instructions. This objective, rather than imparting new knowledge, minimizes the impact of potential domain gaps in the SFT dataset. Consequently, the choice of the SFT dataset is less critical, as long as it provides sufficient examples to establish the desired response format.
>
> - Our SFT model, trained on domain-specific datasets like FGVC-Aircraft and Stanford-Cars, has been evaluated on diverse benchmarks with significant domain gaps, such as Bird-200 in Table 1 and LVIS in Table 3. Despite these challenges, our method consistently outperforms baselines, demonstrating its robustness to catastrophic forgetting.

---

> ### Author Response · Authors · 2024-11-26
>
> Dear Reviewer 5xbU,
>
> Thank you again for your time and valuable feedback on our paper. We have carefully addressed all your comments in our rebuttal and would greatly appreciate it if you could review our response and consider adjusting your score accordingly.
>
> Your support means a lot to us, and we are grateful for your consideration.
>
> Best regards,
>
> Authors

---

> ### Author Response · Authors · 2024-11-27
>
> Dear Reviewer 5xbU,
>
> I hope this email finds you well. I’m writing to kindly follow up on the review of our paper.
>
> We are encouraged to share that one of the reviewers, nzRY, has maintained their original score of 6 and increased the confidence level to 4, reflecting their strengthened belief in the quality and significance of our work.
>
> We sincerely value your feedback and would greatly appreciate it if you could share your comments and scoring at your earliest convenience. Your insights are critical for the timely progress of the review process.
>
> Thank you very much for your time and support.
>
> Best regards,
>
> Authors

---

> ### Author Response · Authors · 2024-12-01
>
> Dear Reviewer 5xbU.
>
> The deadline is approaching. Now two reviewers post their response to our rebuttal, and our paper received mixed results (one 6 rating and one 5 rating). We have not yet received a response from you. Could you please review our rebuttal and share your comments? Thank you.

---

> > ### Author Response · Authors · 2024-12-03
> >
> > Dear Reviewer 5xbU.
> >
> > The deadline is fast approaching, and we’ve already received mixed feedback from two reviewers (one giving a rating of 6 and the other 5). However, we have not yet received your response. With only a few hours left, could you please take a moment to review our rebuttal and provide your comments? Your input would be greatly appreciated. Thank you!

---

> ### Comment · Reviewer_5xbU · 2024-12-03
>
> I have carefully read the author's response, which addressed some of my questions. Overall, I still maintain my original rating.

---

> > ### Author Response · Authors · 2024-12-03
> >
> > Dear Reviewer 5xbU，
> >
> > Thank you very much for your response. We truly appreciate your careful consideration of our rebuttal. We will revise the manuscript thoroughly based on the reviewers' feedback. Given the mixed results with two ratings of 6 and one of 5, we would be grateful if you could clarify any remaining concerns that we may not have fully addressed. Your further insights would be extremely valuable in helping us refine the paper and ensure we have covered all important points. Thank you again for your time and thoughtful feedback.
> >
> > With sincere appreciation,
> > The Authors

---

### Meta-Review · Area_Chair_WuL7 · 2024-12-25

**Metareview:**

The submission addresses the problem of few-shot / zero-shot image classification and object detection with vision-language models. It proposes to combine the strengths of contrastive language-image pre-trained models (CLIP) and multimodal large language models (MLLMs). This is achieved by its proposed "retrieving and ranking augmented" MLLMs. The submission received mixed ratings after rebuttal, including two borderline accepts (6) and two borderline rejects (5). Below I summarize the main strengths and limitations of the submission, according to the reviews (after rebuttal discussion) and my own reading of the submission:

*Strengths:*
- The proposed framework is simple (thus easy to re-implement) and effective (as demonstrated on multiple image classification and detection benchmarks).

*Weaknesses:*
- The introduced method appears to have limited "technical novelty" (nzRY, FduB, 5xbU).

Specifically, after rebuttal, all questions from nzRY have been addressed, while FduB voiced their shared concern with 5xbU that the MLLM is only used for re-ranking final predictions, and has limited "novelty". Reviewer 5xbU acknowledged the rebuttal but did not specify which of their questions are yet to be resolved, and Zt9H did not engage in the rebuttal. After reading the submission and the discussions, the AC has the following opinions:
- The AC disagrees with the authors' characterization of "RAG" as solely for text-generation and for NLP. Prior work on multimodal RAG includes "Retrieval-Augmented Multimodal Language Modeling" (ICML 2023, cited by the submission itself), "REVEAL: Retrieval-Augmented Visual-Language Pre-Training with Multi-Source Multimodal Knowledge Memory" (CVPR 2023), among many others. As such, the authors' argument on "novelty" from the perspective of "RAR is fundamentally different from RAG" is not convincing to the AC.
- Reviewer 5xbU's question about the necessity of using *large* models was not addressed by the rebuttal.
- Reviewer Zt9H's questions appear to be mostly addressed by the authors.
- The authors cited "Novelty in Science" in their rebuttal. While the AC enjoyed the cited article since it was published, the AC respectfully disagrees with the authors' argument in the context of this submission. Specifically, while the narrow definition of "technical novelty" is not always required for a high-quality ICLR paper, the AC has reservations on: (1) The justification of combining "large models" (5xbU) into solving specialized tasks. In the AC's humble opinon, the simplicity should refer not only to the design when combining different modules, but also to the simplicity and efficiency of the overall framework, and to be aware of the potential overhead introduced by additional modules, especially when the module is a "foundation model"; (2) How widely applicable the observations are. This would be particularly relevant since MLLMs are often evaluated not only for recognition (the submission's focus), but also on broader range of understanding and reasoning tasks.

Due to the rationale above, the AC believes that the submission is not ready to be accepted by ICLR 2025.

**Additional Comments On Reviewer Discussion:**

Please find the AC's discussions above.

---

### Decision · Program_Chairs · 2025-01-22

Reject